# MADiff: Offline Multi-agent Learning with Diffusion Models

## Abstract

Diffusion model (DM), as a powerful generative model, recently achieved huge success in various scenarios including offline reinforcement learning, where the policy learns to conduct planning by generating trajectory in the online evaluation. However, despite the effectiveness shown for single-agent learning, it remains unclear how DMs can operate in multi-agent problems, where agents can hardly complete teamwork without good coordination by independently modeling each agent's trajectories. In this paper, we propose MADiff, a novel generative multi-agent learning framework to tackle this problem. MADiff is realized with an attention-based diffusion model to model the complex coordination among behaviors of multiple diffusion agents. To the best of our knowledge, MADiff is the first diffusion-based multi-agent offline RL framework, which behaves as both a decentralized policy and a centralized controller. During decentralized executions, MADiff simultaneously performs teammate modeling, and the centralized controller can also be applied in multi-agent trajectory predictions. Our experiments show the superior performance of MADiff compared to baseline algorithms in a wide range of multi-agent learning tasks, which emphasizes the effectiveness of MADiff in modeling complex multi-agent interactions.

## 1 Introduction

Diffusion models (DMs) (Song & Ermon, 2019; Ho et al., 2020) have achieved remarkable success in various generative modeling tasks (Song et al., 2020; Meng et al., 2021a; Xu et al., 2022), owing to their exceptional abilities at capturing complex data distributions. Thus, some recent work is inspired to build more powerful decision models by introducing DMs into offline reinforcement learning (RL) (Janner et al., 2022; Ajay et al., 2023). Offline RL problems present significant challenges as they prohibit interactions with environments. Consequently, RL algorithms using temporal difference learning may incur severe extrapolation errors due to distributional shift between the behavior policy in the dataset and the learned policy (Fujimoto et al., 2019). This means value functions may be updated using out-of-distribution data points. While previous methods have adopted various regularization techniques on either policies or value functions to mitigate this issue, extrapolation errors persist. Alternatively, viewing offline RL as return-conditioned trajectory generation with DMs allows us to avoid extrapolation errors and leverage the advances of DMs. Several offline RL approaches have been proposed to make use of DMs for planning by generating trajectories conditioned on a high return during online interactions, which also preserves the flexibility of combining skills and constraints into trajectory generation (Janner et al., 2022; Ajay et al., 2023).

Despite its effectiveness in single-agent learning, applying the generative framework to multi-agent (MA) decision tasks remains uncertain. This is due to the need for modeling interactions and coordination among agents, while each agent still make individual decisions in a decentralized manner. Consequently, merely adopting existing diffusion RL methods by treating each agent's policies independently can result in serious inconsistencies in behavior due to a lack of proper credit assignment among agents. Current multi-agent learning (MAL) approaches typically train a centralized value function to update individual agents' policies (Rashid et al., 2020; Foerster et al., 2018) or use an autoregressive transformer to determine each agent's actions (Meng et al., 2021b; Wen et al., 2022). However, when online interactions are unavailable, learning an incorrect centralized value for each agent can lead to significant extrapolation errors (Fujimoto et al., 2019), and the transformer can only serve as an independent model for each agent.

In this paper, we aim to study the potential of employing DMs to solve the above challenges in offline MAL problems. In particular, we formulate MAL as conditional generative modeling problems, where the target is to generate each agent's trajectory, conditioned on the information of all agents (centralized case) or each agent (decentralized case). To address the aforementioned coordination challenges, we propose the first centralized-training-decentralized-execution (CTDE) diffusion framework for MA problems, named MADIFF. MADIFF adopts a novel attention-based DM to learn a return-conditional trajectory generation model on a reward-labeled multi-agent interaction dataset. In particular, the designed attention is computed in several latent layers of the model of each agent to fully interchange the information and integrate the global information of all agents. To model the coordination among agents, MADIFF applies the attention mechanism on latent embedding for information interaction across agents. During training, MADIFF performs centralized training on the joint trajectory distributions of all agents from offline datasets, including different levels of expected returns. During inference, MADIFF adopts classifier-free guidance with low-temperature sampling to generate behaviors given the conditioned high expected returns, allowing for decentralized execution by predicting the behavior of other agents and generating its own behavior. Therefore, MADIFF can be regarded as a principled offline MAL solution that not only serves as a decentralized policy for each agent or a centralized controller for all agents, but also includes teammate modeling without additional cost. Comprehensive experiments demonstrated superior performances of MADIFF on various multi-agent learning tasks, including offline RL and trajectory prediction.

In summary, our contributions are (1) a novel attention-based DM structure that is designed explicitly for MAL and enables coordination among agents in each denoising step; (2) a principled MAL solution that unifies decentralized policy, centralized controller, teammate modeling, and trajectory prediction; (3) achieving superior performances for various offline multi-agent problems.

## 2 PRELIMINARIES

### 2.1 MULTI-AGENT OFFLINE REINFORCEMENT LEARNING

We consider a partially observable and fully cooperative multi-agent learning (MAL) problem, where agents with local observations cooperate to finish the task. Formally, it is defined as a Dec-POMDP (Oliehoek & Amato, 2016): $G = \langle \mathcal{S}, \mathcal{A}, P, r, \Omega, O, N, U, \gamma \rangle$, where $\mathcal{S}$ and $\mathcal{A}$ denote state and action space separately, and $\gamma$ is the discounted factor. The system includes $N$ agents $\{1, 2, \ldots, N\}$ act in discrete time steps, and starts with an initial global state $s_0 \in \mathcal{S}$ sampled from the distribution $U$. At each time step $t$, every agent $i$ only observes a local observation $o^i \in \Omega$ produced by the function $O(s, a) : \mathcal{S} \times \mathcal{A} \to \Omega$ and decides $a \in \mathcal{A}$, which forms the joint action $\mathbf{a} \in \mathcal{A} \equiv \mathcal{A}^N$, leading the system transits to the next state $s'$ according to the dynamics function $P(s'|s, \mathbf{a}) : \mathcal{S} \times \mathcal{A} \to \mathcal{S}$. Normally, agents receive a shared reward $r(s, \mathbf{a})$ at each step, and the optimization objective is to learn a policy $\pi^i$ for each agent that maximizes the discounted cumulative reward $\mathbb{E}_{s_t, \mathbf{a}_t}[\sum_t \gamma^t r(s_t, \mathbf{a}_t)]$. In offline settings, instead of collecting online data in environments, we only have access to a static dataset $\mathcal{D}$ to learn the policies. The dataset $\mathcal{D}$ is generally composed of trajectories $\boldsymbol{\tau}$, *i.e.*, observation-action sequences $[\boldsymbol{o_0}, \boldsymbol{a_0}, \boldsymbol{o_1}, \boldsymbol{a_1}, \cdots, \boldsymbol{o_T}, \boldsymbol{a_T}]$ or observation sequences $[\boldsymbol{o_0}, \boldsymbol{o_1}, \cdots, \boldsymbol{o_T}]$. We use bold symbols to denote the joint vectors of all agents.

### 2.2 DIFFUSION PROBABILISTIC MODELS

Diffusion models (DMs) (Sohl-Dickstein et al., 2015; Song & Ermon, 2019; Ho et al., 2020), as a powerful class of generative models, implement the data generation process as reversing a forward noising process (denoising process). For each data point $x_0 \sim p_{\text{data}}(x)$ from the dataset $\mathcal{D}$, the noising process is a discrete Markov chain $x_{0:K}$ such that $p(x_k|x_{k-1}) = \mathcal{N}(x_k|\sqrt{\alpha_k}x_{k-1}, (1 - \alpha_k)I)$, where $\mathcal{N}(\mu, \Sigma)$ denotes a Gaussian distribution with mean $\mu$ and variance $\Sigma$, and $\alpha_{0:K} \in \mathbb{R}$ are hyperparameters which control the variance schedule. The variational reverse Markov chain is parameterized with $q_\theta(x_{k-1}|x_k) = \mathcal{N}(x_{k-1}|\mu_\theta(x_k, k), (1 - \alpha_k)I)$. The data sampling process begins by sampling an initial noise $x_K \sim \mathcal{N}(0, I)$, and follows the reverse process until $x_0$. The reverse process can be estimated by optimizing a simplified surrogate loss as in Ho et al. (2020):

$$\mathcal{L}(\theta) = \mathbb{E}_{k \sim [1,K], x_0 \sim q, \epsilon \sim \mathcal{N}(0,I)} \left[ \|\epsilon - \epsilon_\theta (x_k, k)\|^2 \right] . \tag{1}$$

The estimated mean of Gaussian can be written as $\mu_\theta(x_k, k) = \frac{1}{\sqrt{\alpha_k}} \left( x_k - \frac{1-\alpha_k}{\sqrt{1-\bar{\alpha}_k}} \epsilon_\theta(x_k, k) \right)$, where $\bar{\alpha}_k = \Pi_{s=1}^k \alpha_s$.

## 2.3 DIFFUSING DECISION MAKING

We introduce how DMs are adopted in resolving single-agent decision-making problems.

**Diffusing over state trajectories and planning with inverse dynamics model.** The form of data that DMs should train on is a critical choice for model usage. Among existing works in single-agent learning, Janner et al. (2022) chose to diffuse over state-action sequences, so that the generated actions for the current step can be directly used for executing. Another choice is diffusing over state trajectories only (Ajay et al., 2023), which is claimed to be easier to model and can obtain better performance due to the less smooth nature of action sequences:

$$\hat{\tau} := [s_t, \hat{s}_{t+1}, \cdots, \hat{s}_{t+H-1}], \tag{2}$$

where $t$ is the sampled time step and $H$ denotes the trajectory length (horizon) modelled by DMs. But only diffusing of states can not provide actions to be executed during online evaluation. Therefore, an additional inverse dynamics model is trained to predict the action $\hat{a}_t$ that makes the state transit from $s_t$ to the generated next state $\hat{s}_{t+1}$:

$$\hat{a}_t = I_\phi(s_t, \hat{s}_{t+1}). \tag{3}$$

Therefore, at every environment step $t$, the agent first plans the state trajectories using an offline-trained DM, and infers the action with the inverse dynamics model.

**Classifier-free guided generation.** For targeted behavior synthesis, DMs should be able to generate future trajectories by conditioning the diffusion process on an observed state $s_t$ and information $y$. We use classifier-free guidance (Ho & Salimans, 2022) which requires taking $y(\tau)$ as additional inputs for the diffusion model. Formally, the sampling procedure starts with Gaussian noise $\hat{\tau}_K \sim \mathcal{N}(0, \alpha I)$, and diffuse $\hat{\tau}_k$ into $\hat{\tau}_{k-1}$ at each diffusion step $k$. Here $\alpha \in [0, 1)$ is the scaling factor used in low-temperature sampling to scale down the variance of initial samples (Ajay et al., 2023). We use $\tilde{x}_{k,t}$ to denote the denoised state $s_t$ at $k$'s diffusion step. $\hat{\tau}_k$ denotes the denoised trajectory at $k$'s diffusion step for a single agent: $\hat{\tau}_k := [s_t, \tilde{x}_{k,t+1}, \cdots, \tilde{x}_{k,t+H-1}]$. Note that for sampling during evaluations, the first state of the trajectory is always set to the current observed state at all diffusion steps for conditioning, and every diffusion step proceeds with the perturbed noise:

$$\hat{\epsilon} := \epsilon_\theta(\tau_k, \emptyset, k) + \omega(\tau, w(y(\tau), k) - \epsilon_\theta(\hat{\tau}_k, \emptyset, k)), \tag{4}$$

where $\omega$ is a scalar for extracting the distinct portions of data with characteristic $y(\tau)$. By iterative diffusing the noisy samples, we can obtain a clean state trajectory: $\hat{\tau}_0(\tau) := [s_t, \hat{s}_{t+1}, \cdots, \hat{s}_{t+H-1}]$.

## 3 METHODOLOGY

We formulate the problem of MAL as conditional generative modeling:

$$\max_\theta \mathbb{E}_{\boldsymbol{\tau} \sim \mathcal{D}}[\log p_\theta(\boldsymbol{\tau}|\boldsymbol{y}(\cdot))], \tag{5}$$

where $p_\theta$ is learned for estimating the conditional data distribution of joint trajectory $\boldsymbol{\tau}$, given information $\boldsymbol{y}(\cdot)$, such as observations, rewards, and constraints. When all agents are managed by a centralized controller, *i.e.*, the decisions of all agents are made jointly, we can learn the generative model by conditioning the global information aggregated from all agents $\boldsymbol{y}(\boldsymbol{\tau})$; otherwise, if we consider each agent $i$ separately and require each agent to make decisions in a decentralized manner, we can only utilize the local information $y^i(\tau^i)$ of each agent $i$, including the private information and the common information shared by all (*e.g.*, team rewards).

### 3.1 MULTI-AGENT DIFFUSION WITH ATTENTION

In order to handle MAL problems, agents must learn to coordinate. To solve the challenge of modeling the complex inter-agent coordination in the dataset, we propose a novel attention-based diffusion architecture designed to fully interchange information among agents.

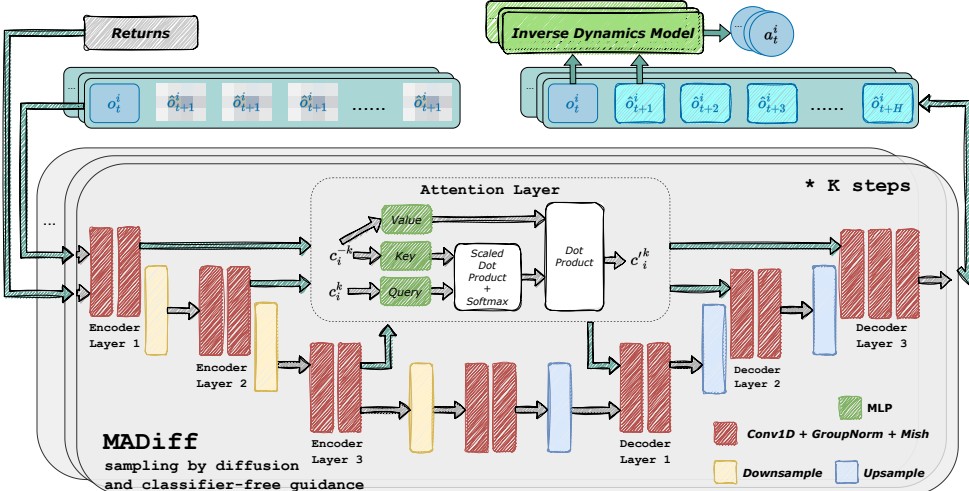

Figure 1: The architecture of MADIFF, which is an attention-based diffusion network framework that performs attention across all agents at every decoder layer of each agent. "Returns" is set to the cumulative discounted reward starting from time step $t$.

The overall architecture is illustrated in Fig. 1. In detail, we adopt U-Net as the base structure for modeling agents' individual trajectories, which consists of repeated one-dimensional convolutional residual blocks. Afterward, to encourage information interchange and improve coordination ability, a critical change is made by adopting attention (Vaswani et al., 2017) layers before all decoder blocks in the U-Nets of all agents. Besides, since embedding vectors from different agents are aggregated by the attention operation rather than concatenations, MADIFF is index-free such that the input order of agents can be arbitrary and does not affect the results.

Formally, the input to $l$-th decoder layer in the U-Net of each agent $i$ is composed of two components: the skip-connected feature $c_l^i$ from the symmetric $l$-th encoder layer and the embedding $e_l^i$ from the previous decoder layer. The computation of attention in MADIFF is conducted on $c_l^i$ rather than $e_l^i$ since in the U-Net structure the encoder layers are supposed to extract informative features from the input data. We adopt the multi-head attention mechanism (Vaswani et al., 2017) to fuse the encoded feature $c_i'^k$ with other agents' information, which is important in effective multi-agent coordination.

## 3.2 CENTRALIZED TRAINING OBJECTIVES

Given an offline dataset $\mathcal{D}$ that consists of all agents' trajectories data with rewards or other information provided, we train MADIFF which is parameterized through the unified noise model $\epsilon_\theta$ for all agents and the inverse dynamics model $I_\phi^i$ of each agent $i$ with the reverse diffusion loss and the inverse dynamics loss:

$$\mathcal{L}(\theta, \phi) := \mathbb{E}_{k, \boldsymbol{\tau}_0 \in \mathcal{D}, \beta \sim \text{Bern}(p)}[\|\epsilon - \epsilon_\theta(\hat{\boldsymbol{\tau}}_k, (1 - \beta)\boldsymbol{y}(\boldsymbol{\tau}_0) + \beta\emptyset, k)\|^2]$$
$$+ \sum_i \mathbb{E}_{(s^i, a^i, s'^i) \in \mathcal{D}}[\|a^i - I_\phi^i(o^i, o'^i)\|^2] . \tag{6}$$

For training the DM, we sample noise $\epsilon \sim \mathcal{N}(\mathbf{0}, \boldsymbol{I})$ and a time step $k \sim \mathcal{U}\{1, \cdots, K\}$, construct a noise corrupted joint state sequence $\boldsymbol{\tau}_k$ from $\boldsymbol{\tau}$ and predict the noise $\hat{\epsilon}_\theta := \epsilon_\theta(\hat{\boldsymbol{\tau}}_k, \boldsymbol{y}(\boldsymbol{\tau}_0), k)$. Note that the noisy array $\hat{\boldsymbol{\tau}}_k$ is applied with the same condition required by the sampling process, as we will discuss in Section 3.3 in detail. In addition, the training of MADIFF is centralized such that all agents' data is fully utilized, and the attention is conducted among all agents. As for the inverse dynamics training, we sample the observation transitions of each agent to predict the action.

It is worth noting that the choice of whether agents should share their parameters of $\epsilon_\theta^i$ and $I_{\phi^i}$ depending on the homogeneous nature and requirements of tasks. If agents choose to share their parameters, only one shared DM and inverse dynamics model are used for generating all agents' trajectories; otherwise, each agent $i$ has extra parameters (i.e., the U-Net and inverse dynamic models) to generate their states and predict their actions. The attention modules are always shared to incorporate global information into generating each agent's trajectory.

### 3.3 CENTRALIZED CONTROL OR DECENTRALIZED EXECUTION

**Centralized control.** A direct and straightforward way to utilize MADIFF in online decision-making tasks is to have a centralized controller for all agents. The centralized controller has access to all agents' current local observations and generates all agents' trajectories along with predicting their actions, which are sent to every single agent for acting in the environment. This is applicable for multi-agent trajectory prediction problems and when interactive agents are permitted to be centralized controlled, such as in team games. During the generation process, we sample an initial noise trajectory $\hat{\boldsymbol{\tau}}_K$, condition the current joint states of all agents and the global information to utilize $\boldsymbol{y}(\boldsymbol{\tau}_0)$; following the diffusion step described in Eq. (4) with $\epsilon_\theta$, we finally sample the joint observation sequence $\hat{\boldsymbol{\tau}}_0$ as below:

$$\underbrace{\left[\boldsymbol{o}_t, \tilde{\boldsymbol{x}}_{K,t+1}, \cdots, \tilde{\boldsymbol{x}}_{K,t+H-1}\right]}_{\hat{\boldsymbol{\tau}}_K} \xrightarrow{\text{Iterative } K \text{ diffusion steps}} \underbrace{\left[\boldsymbol{o}_t, \hat{\boldsymbol{o}}_{t+1}, \cdots, \hat{\boldsymbol{o}}_{t+H-1}\right]}_{\hat{\boldsymbol{\tau}}_0} , \tag{7}$$

where every $\tilde{\boldsymbol{x}}_t \sim \mathcal{N}(\boldsymbol{0}, \boldsymbol{I})$ is a noise vector sampled from the normal Gaussian. After generation, each agent obtains the action through its own inverse dynamics model following Eq. (3) using the current observation $o_t^i$ and the predicted next observation $\hat{o}_{t+1}^i$, and takes a step in the environment. We highlight that MADIFF provides an efficient way to generate joint actions and the attention module guarantees sufficient feature interactions and information interchange among agents.

**Decentralized execution with teammate modeling.** Compared with centralized control, a more popular and widely-adopted setting is that each agent only makes its own decision without any communication with other agents, which is what most current works (Lowe et al., 2017; Rashid et al., 2020; Pan et al., 2022) dealt with. In this case, we can only utilize the current local observation of each agent $i$ to plan its own trajectory. To this end, the initial noisy trajectory is conditioned on the current observation of the agent $i$. Similar to the centralized case, by iterative diffusion steps, we finally sample the joint state sequence based on the local observation of agent $i$ as:

$$\underbrace{\begin{bmatrix} \tilde{x}_{K,t}^0, \cdots, \tilde{x}_{K,t+H-1}^0 \\ \cdots, \\ o_t^i, \cdots, \tilde{x}_{K,t+H-1}^i \\ \cdots, \\ \tilde{x}_{K,t}^N, \cdots, \tilde{x}_{K,t+H-1}^N \end{bmatrix}}_{\hat{\boldsymbol{\tau}}_K^i} \xrightarrow{\text{Iterative } K \text{ diffusion steps}} \underbrace{\begin{bmatrix} \hat{o}_t^0, \cdots, \hat{o}_{t+H-1}^0 \\ \cdots, \\ o_t^i, \cdots, \hat{o}_{t+H-1}^i \\ \cdots, \\ \hat{o}_t^N, \cdots, \hat{o}_{t+H-1}^N \end{bmatrix}}_{\hat{\boldsymbol{\tau}}_0^i} , \tag{8}$$

and we can also obtain the action through the agent $i$'s inverse dynamics model as mentioned above. An important observation is that, the decentralized execution of MADIFF includes teammate modeling such that the agent $i$ infers all others' observation sequences based on its own local observation. We show in experiments that this achieves great performances in various tasks, indicating the effectiveness of teammate modeling and the great ability in coordination.

**History-based generation.** We find DMs are good at modeling the long-term joint distributions, and as a result MADIFF perform better in some cases when we choose to condition on the trajectory of the past history instead of only the current observation. This implies that we replace the joint observation $\boldsymbol{o}_t$ in Eq. (7) as the $C$-length joint history sequence $\boldsymbol{h}_t := [\boldsymbol{o}_{t-C}, \cdots, \boldsymbol{o}_{t-1}, \boldsymbol{o}_t]$, and replace the independent observation $o_t^i$ in Eq. (8) as the history sequence $h_t^i := [o_{t-C}^i, \cdots, o_{t-1}^i, o_t^i]$ of each agent $i$. An illustration of how agents' history and future trajectories are modeled by MADIFF in both centralized control and decentralized execution is provided in the Appendix Section B.

## 4 RELATED WORK

**Diffusion Models for Decision Making.** There is a recent line of work applying diffusion models (DMs) to decision-making problems such as RL and imitation learning. Janner et al. (2022) demonstrate the potential of diffusion models for offline RL, by designing a diffusion-based trajectory generation model and training a value function to sample high-rewarded trajectories. A consequent work (Ajay et al., 2023) takes conditions as inputs to the DM, thus bringing more flexibility that generates behaviors that satisfy combinations of diverse conditions. Different from above, Diffusion-QL (Wang et al., 2022) instead uses the DM as a form of policy, *i.e.*, generating actions conditioned on

states, and the training objective behaves as a regularization under the framework of TD-based offline RL algorithms. All of these existing methods focus on solving single-agent tasks. The proposed MADIFF is structurally similar to Ajay et al. (2023), but includes effective modules to model agent coordination in MAL tasks.

**Multi-agent Offline RL.** While offline RL has become an active research topic, only a limited number of works studied offline MARL due to the challenge of offline coordination. Jiang & Lu (2021) extended BCQ (Fujimoto et al., 2019), a single-agent offline RL algorithm with policy regularization to multi-agent; Yang et al. (2021) developed an implicit constraint approach for offline Q learning, which was found to perform particularly well in MAL tasks; Pan et al. (2022) argued the actor update tends to be trapped in local minima when the number of agents increases, and correspondingly proposed an actor regularization method named OMAR. All of these Q-learning-based methods naturally have extrapolation error problem (Fujimoto et al., 2019) in offline settings, and their solution cannot get rid of it but only mitigate some. As an alternative, MADT (Meng et al., 2021b) formulated offline MARL as return-conditioned supervised learning, and use a similar structure to a previous transformer-based offline RL work (Chen et al., 2021). However, offline MADT learns an independent model for each agent without modeling agent interactions; it relies on the gradient from centralized critics during online fine-tuning to integrate global information into each agent's decentralized policy. To the best of our knowledge, MADIFF is the first work to investigate the usage of DMs in MAL. MADIFF not only avoids the problem of extrapolation error, but also achieves the modeling of collaborative information while allowing CTDE in a completely offline training manner.

**Opponent Modeling in MARL.** Our modeling of teammates can be placed under the larger framework of opponent modeling, which refers to the process by which an agent tries to infer the behaviors or intentions of other agents using its local information. There is a rich literature on utilizing opponent modeling in online MARL. Rabinowitz et al. (2018) uses meta-learning to build three models, and can adapt to new agents after observing their behavior. SOM (Raileanu et al., 2018) uses the agent's own goal-conditioned policy to infer other agents' goals from a maximum likelihood perspective. LIAM (Papoudakis et al., 2021) extract representations of other agents with variational auto-encoders conditioned on the controlled agent's local observations. Considering the impact of the ego agent's policy on other agents' policies, LOLA (Foerster et al., 2017) and following works (Willi et al., 2022; Zhao et al., 2022) instead model the parameter update of the opponents. Different from these methods, MADIFF-D uses the same generative model to jointly output plans of its own trajectory and predictions of other agents' trajectories and is shown to be effective in offline settings.

## 5 EXPERIMENTS

In experiments, we are aiming at excavating the ability of MADIFF in modeling the complex interactions among cooperative agents, particularly, whether MADIFF is able to (i) generate high-quality multi-agent trajectories; (ii) appropriately infer teammates' behavior and (iii) learn effective, coordinated policies from offline data.

### 5.1 TASK DESCRIPTIONS

To fully test MADIFF, we conduct experiments on multiple commonly used multi-agent testbeds, including three offline MARL challenges and a multi-agent trajectory prediction benchmark.

- **Multi-agent particle environments (MPE)** (Lowe et al., 2017): multiple particles must cooperate in a 2D plane to achieve a common goal. Three tasks are adopted from MPE. *Spread*, three agents start at some random locations and have to cover three landmarks without collisions; *Tag*, three predictors try to catch a pre-trained prey opponent that moves faster and needs cooperative containment; *World*, also requires three predators to catch a pre-trained prey, whose goal is to eat the food on the map while not getting caught, and the map has forests that agents can hide and invisible from the outside.
    - **Datasets**: we use the offline datasets constructed by Pan et al. (2022), including four datasets collected by policies of different qualities trained by MATD3 (Ackermann et al., 2019), namely, Expert, Medium-Replay (Md-Replay), Medium and Random, depending on the quality of the policy used to collect the data. They also add noises to increase data diversity.

- **Multi-Agent Mujoco (MA Mujoco)** (Peng et al., 2021): different subsets of a robot's joints are controlled by independent agents, and the goal is to make the robot run forward as fast as possible. We use two configurations: *2-agent halfcheetah (2halfcheetah)* and *4-agent ant (4ant)*.
  - **Datasets**: we use the off-the-grid offline dataset (Formanek et al., 2023), including three datasets with different qualities for each robot control task, *e.g.*, Good, Medium, and Poor.
- **StarCraft Multi-Agent Challenge (SMAC)** (Samvelyan et al., 2019): a team of either homogeneous or heterogeneous units collaborates to fight against the enemy team that is controlled by the hand-coded built-in StarCraft II AI. We cover two maps: *3m*, both teams have three Marines; and *5m_vs_6m (5m6m)*, requires controlling five Marines and the enemy team has six Marines.
  - **Datasets**: we use the off-the-grid offline dataset (Formanek et al., 2023), including three datasets with different qualities for each map, *e.g.*, Good, Medium, and Poor.
- **Multi-Agent Trajectory Prediction (MATP)**: different from the former offline MARL challenges which should learn the policy for each agent, the MATP problem only requires predicting the future behaviors of all agents, and no decentralized model is needed.
  - **NBA dataset**: the dataset consists of various basketball players recorded trajectories from 631 games in the 2015-2016 season. Following Alcorn & Nguyen (2021), we split 569/30/32 training/validation/test games, with downsampling from the original 25 Hz to 5Hz. Different from MARL tasks, other information apart from each player's historical trajectories is available for making predictions, including the ball's historical trajectories, player ids, and a binary variable indicating the side of each player's frontcourt. Each term is encoded using embedding networks and concatenated with diffusion time embeddings as side inputs to each U-Net block.

## 5.2 COMPARED BASELINES AND METRICS

For offline MARL environments, we use the episodic return obtained in online rollout as the performance measure. We include MA-ICQ (Yang et al., 2021) and MA-CQL (Kumar et al., 2020) as baselines on all offline RL tasks. On MPE, we also include OMAR and MA-TD3+BC (Fujimoto & Gu, 2021) in baseline algorithms and use the results reported by Pan et al. (2022). On MA Mujoco, baseline results are adopted from Formanek et al. (2023). On SMAC, we include MADT (Meng et al., 2021b) and other baselines benchmarked by Formanek et al. (2023), which also covers an offline learning version of QMIX (Rashid et al., 2020). In addition, we implement independent behavior cloning (BC) as a naive supervised learning baseline for comparison.

For MATP problem, we use distance-based metrics including average displacement error (ADE) $\frac{1}{L \cdot N} \sum_{t=1}^{L} \sum_{i=1}^{N} \|\hat{o}_t^i - o_t^i\|$ and final displacement error (FDE) $\frac{1}{N} \sum_{i=1}^{N} \|\hat{o}_L^i - o_L^i\|$, where $L$ is the prediction length (Li et al., 2020). We also report minADE$_{20}$ and minFDE$_{20}$ as additional metrics to balance the stochasticity in sampling, which are the minimum ADE and FDE among 20 predicted trajectories, respectively. We compare MADIFF with Baller2Vec++ (Alcorn & Nguyen, 2021), an autoregressive MATP algorithm based on transformer and specifically designed for the NBA dataset.

## 5.3 NUMERICAL RESULTS

We reported the numerical results both for the CTDE version of MADIFF (denoted as MADIFF-D) and the centralized version MADIFF (MADIFF-C). For offline MARL, since baselines are tested in a decentralized style, *i.e.*, all agents independently decide their actions with only local observations, MADIFF-C is not meant to be a fair comparison but to show if MADIFF-D fills the gap for coordination without global information. For MATP, due to its centralized prediction nature, MADIFF-C is the only variant involved.

**Offline MARL.** As listed in Tab. 1, MADIFF-D achieves the best result on most of the datasets. Similar to the single-agent case, direct supervised learning (BC) on the dataset behaves poorly when datasets are mixed quality. Offline RL algorithms such as MA-CQL that compute conservative values have a relatively large drop in performance when the dataset quality is low. Part of the reason may come from the fact that those algorithms are more likely to fall into local optima in multi-agent scenarios (Pan et al., 2022). Thanks to the distributional modeling ability of the DM, MADIFF-D generally obtains better or competitive performance compared with OMAR (Pan et al., 2022) without any design for avoiding bad local optima similar to Pan et al. (2022). On SMAC tasks, MADIFF-D achieves comparable performances, although it is slightly degraded compared with MADIFF-C.

Table 1: The average score on MPE and SMAC tasks. Shaded columns represent our methods. The mean and standard error are computed over 3 different seeds.

| Testbed | Dataset | Task | BC | QMIX | MA-ICQ | MA-TD3+BC | MA-CQL | OMAR | MADT | MADiff-D | MADiff-C |
|---|---|---|---|---|---|---|---|---|---|---|---|
| MPE | Expert | Spread | 35.0 ± 2.6 | - | 104.0 ± 3.4 | 108.3 ± 3.3 | 98.2 ± 5.2 | 114.9 ± 2.6 | - | 98.4 ± 12.7 | 114.7 ± 5.3 |
| | | Tag | 40.0 ± 9.6 | - | 113.0 ± 14.4 | 115.2 ± 12.5 | 93.9 ± 14.0 | 116.2 ± 19.8 | - | 163.7 ± 16.3 | 146.6 ± 12.8 |
| | | World | 33.0 ± 9.9 | - | 109.5 ± 22.8 | 110.3 ± 21.3 | 71.9 ± 28.1 | 110.4 ± 25.7 | - | 112.9 ± 22.1 | 170.7 ± 10.3 |
| | Md-Replay | Spread | 10.0 ± 3.8 | - | 13.6 ± 5.7 | 15.4 ± 5.6 | 20.0 ± 8.4 | 37.9 ± 12.3 | - | 42.9 ± 11.6 | 47.2 ± 6.6 |
| | | Tag | 0.9 ± 1.4 | - | 34.5 ± 27.8 | 28.7 ± 20.9 | 24.8 ± 17.3 | 47.1 ± 15.3 | - | 83.9 ± 9.8 | 74.2 ± 13.2 |
| | | World | 2.3 ± 1.5 | - | 12.0 ± 9.1 | 17.4 ± 8.1 | 29.6 ± 13.8 | 42.9 ± 19.5 | - | 39.4 ± 21.1 | 86.5 ± 10.2 |
| | Medium | Spread | 31.6 ± 4.8 | - | 29.3 ± 5.5 | 29.3 ± 4.8 | 34.1 ± 7.2 | 47.9 ± 18.9 | - | 53.2 ± 2.3 | 77.9 ± 2.9 |
| | | Tag | 22.5 ± 1.8 | - | 63.3 ± 20.0 | 65.1 ± 29.5 | 61.7 ± 23.1 | 66.7 ± 23.2 | - | 112.5 ± 2.1 | 116.5 ± 12.8 |
| | | World | 25.3 ± 2.0 | - | 71.9 ± 20.0 | 73.4 ± 9.3 | 58.6 ± 11.2 | 74.6 ± 11.5 | - | 121.8 ± 16.0 | 127.5 ± 26.0 |
| | Random | Spread | -0.5 ± 3.2 | - | 6.3 ± 3.5 | 9.8 ± 4.9 | 24.0 ± 9.8 | 34.4 ± 5.3 | - | 19.4 ± 2.9 | 47.1 ± 3.9 |
| | | Tag | 1.2 ± 0.8 | - | 2.2 ± 2.6 | 5.7 ± 3.5 | 5.0 ± 8.2 | 11.1 ± 2.8 | - | 6.7 ± 11.0 | -1.7 ± 2.6 |
| | | World | -2.4 ± 0.5 | - | 1.0 ± 3.2 | 2.8 ± 5.5 | 0.6 ± 2.0 | 5.9 ± 5.2 | - | -4.4 ± 2.1 | 0.1 ± 5.1 |
| MA Mujoco | Good | 2halfcheetah | 6846 ± 574 | - | - | 7025 ± 439 | - | 1434 ± 1903 | - | 8254 ± 179 | 8662 ± 102 |
| | Medium | 2halfcheetah | 1627 ± 187 | - | - | 2561 ± 82 | - | 1892 ± 220 | - | 2215 ± 27 | 2221 ± 56 |
| | Poor | 2halfcheetah | 465 ± 59 | - | - | 736 ± 72 | - | 384 ± 420 | - | 751 ± 74 | 767 ± 42 |
| | Good | 4ant | 2802 ± 133 | - | - | 2628 ± 971 | - | 344 ± 631 | - | 3090 ± 26 | 3087 ± 32 |
| | Medium | 4ant | 1617 ± 153 | - | - | 1843 ± 494 | - | 929 ± 349 | - | 1679 ± 93 | 1897 ± 44 |
| | Poor | 4ant | 1033 ± 122 | - | - | 1075 ± 96 | - | 518 ± 112 | - | 1268 ± 51 | 1332 ± 45 |
| SMAC | Good | 3m | 16.0 ± 1.0 | 13.8 ± 4.5 | 18.8 ± 0.6 | - | 19.6 ± 0.3 | - | 19.0 ± 0.3 | 18.8 ± 0.2 | 19.7 ± 0.1 |
| | Medium | 3m | 8.2 ± 0.8 | 17.3 ± 0.9 | 18.1 ± 0.7 | - | 18.9 ± 0.7 | - | 15.8 ± 0.5 | 17.2 ± 0.3 | 18.4 ± 0.2 |
| | Poor | 3m | 4.4 ± 0.1 | 10.0 ± 2.9 | 14.4 ± 1.2 | - | 5.8 ± 0.4 | - | 4.2 ± 0.1 | 11.2 ± 0.1 | 11.8 ± 1.0 |
| | Good | 5m6m | 16.6 ± 0.6 | 8.0 ± 0.5 | 16.3 ± 0.9 | - | 13.8 ± 3.1 | - | 16.8 ± 0.1 | 17.0 ± 0.6 | 18.1 ± 0.1 |
| | Medium | 5m6m | 12.4 ± 0.9 | 12.0 ± 1.1 | 15.3 ± 0.7 | - | 17.0 ± 1.2 | - | 16.1 ± 0.2 | 17.0 ± 0.2 | 17.6 ± 0.4 |
| | Poor | 5m6m | 7.5 ± 0.2 | 10.7 ± 0.9 | 9.4 ± 0.4 | - | 10.4 ± 1.0 | - | 7.6 ± 0.3 | 10.3 ± 0.5 | 11.0 ± 0.3 |

**MATP on the NBA dataset.** Tab. 2 shows the MATP results on the NBA dataset. As is observed, when comparing ADE and FDE, MADIFF-C significantly outperforms the baseline; however, our algorithm only slightly beats baseline for $minADE_{20}$, and has higher $minFDE_{20}$. We suspect the reason is that Baller2Vec++ has a large prediction variance, as observed in our experiments. When Baller2Vec++ only predicts one trajectory, a few players' trajectories deviate from the truth so far that deteriorate the overall ADE and FDE results, although the prediction of other agents is more accurate. In contrast, when allowing to sample 20 times and calculating the minimum ADE/FDE according to the ground truth, Baller2Vec++ can choose the best trajectory for every single agent, which makes $minADE_{20}$ and $minFDE_{20}$ significantly smaller than one-shot metrics. However, considering it may be not practical to select the best trajectories without access to the ground truth, MADIFF-C is much more stable than Baller2Vec++. Visualizations of predicted trajectories of MADIFF-C and Baller2Vec++ are provided in the Appendix Section F.3.

Table 2: Multi-agent trajectory prediction results on NBA dataset across 3 seeds, given the first step of all agents' positions.

| Traj. Len. | Metric | Baller2Vec++ | MADIFF-C |
|---|---|---|---|
| 20 | ADE | 15.15 ± 0.38 | **7.92 ± 0.86** |
| | FDE | 24.91 ± 0.68 | **14.06 ± 1.16** |
| | $minADE_{20}$ | 5.62 ± 0.05 | **5.20 ± 0.04** |
| | $minFDE_{20}$ | **5.60 ± 0.12** | 7.61 ± 0.19 |
| 64 | ADE | 32.07 ± 1.93 | **17.24 ± 0.80** |
| | FDE | 44.93 ± 3.02 | **26.69 ± 0.13** |
| | $minADE_{20}$ | 14.72 ± 0.53 | **11.40 ± 0.06** |
| | $minFDE_{20}$ | **10.41 ± 0.36** | 11.26 ± 0.26 |

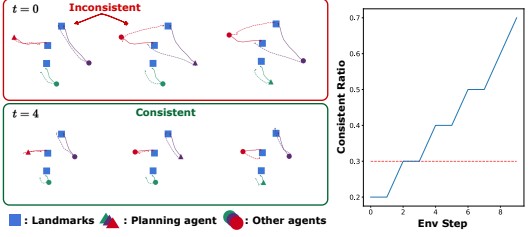

Figure 2: Visualization of an episode in the *Spread* task. The solid lines are real rollout trajectories, and the dashed lines are planned trajectories generated by the DM.

## 5.4 QUALITATIVE ANALYSIS

**Teammate modeling with diffusion models.** In Fig. 2 left, we visualize an episode generated by MADIFF-D trained on the Expert dataset of *Spread* task. The top and bottom rows are snapshots of entities' positions on the initial and intermediate time steps. The three rows from left to right in each column represent the perspectives of the three agents, red, purple, and green, respectively. Dashed lines are the planned trajectories for the controlled agent and other agents output by DMs, and solid lines are the real rollout trajectories. We observe that at the start, the red agent and the purple agent generate *inconsistent* plans, where both agents decide to move towards the middle landmark and assume the other agent is going to the upper landmark. At the intermediate time step, when the red agent is close to the middle landmark while far from the uppermost ones, the purple agent altered the planned trajectories of both itself and the red teammate, which makes all agents' plans *consistent* with each other. This particular case indicates that MADIFF is able to correct the prediction of teammates' behaviors during rollout and modify each agent's own desired goal correspondingly.

In Fig. 2 right, we demonstrate that such corrections of teammate modeling are common and it can help agents make globally coherent behaviors. We sample 100 episodes with different initial states and define *Consistent Ratio* at some time step $t$ as the proportion of episodes in which the three agents make consistent planned trajectories. The horizontal red line represents how many portions of the real rollout trajectories are consistent. The interesting part is that the increasing curve reaches the red line before the final step, and ends up even higher. This indicates that the planned ego and teammates' trajectories are guiding the multi-agent interactions beforehand, which is a strong exemplar of the benefits of MADIFF's teammate modeling abilities. We also include visualizations of imagined teammate observation sequences in SMAC *3m* task in the Appendix Section F.2.

## 5.5 ABLATION STUDY

Our key argument is that the great coordination ability of MADIFF is brought by the attention modules among individual agents' diffusion networks. We validate this insight through a set of ablation experiments on MPE. We compare MADIFF-D with independent DMs, *i.e.*, each agent learns from corresponding offline data using independent U-Nets without attention. We denote this variant as MADIFF-D-Ind. In addition, we also ablate the choice whether each agent should share parameters of their basic U-Net, noted as Share or NoShare. Without causing ambiguity, we omit the name of MADIFF, and notate the different variants as *D-Share*, *D-NoShare*, *Ind-Share*, *Ind-NoShare*.

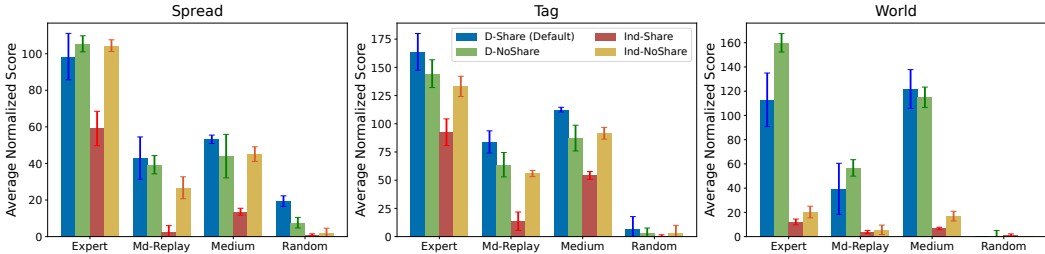

Figure 3: The average normalized score of MADIFF ablation variants in MPE tasks. The mean and standard error are computed over 3 different seeds.

As is obviously observed in Fig. 3, with attention modules, MADIFF-D significantly exceeds that of the independent version on most tasks, justifying the importance of inter-agent attentions. The advantage of MADIFF-D is more evident when the task becomes more challenging and the data becomes more confounded, e.g., results on World, where the gap between centralized and independent models is larger, indicating the difficulty of offline coordination. More importantly, DMs with powerful modeling abilities can more easily discover diverse behaviors in data, which leads to greater difficulty in yielding compatible behavior by independently trained models. As for the parameter sharing choice, the performance of MADIFF-D-Share and MADIFF-D-NoShare is similar overall. Since MADIFF-D-Share has fewer parameters, we prefer MADIFF-D-Share, and we choose it as the default variant compared with other baselines as reported in Tab. 1.

## 6 CONCLUSION AND LIMITATIONS

In this paper, we propose MADIFF, a novel generative multi-agent learning framework, which is realized with an attention-based diffusion model that is designed to model the complex coordination among multiple agents. To our knowledge, MADIFF is the first diffusion-based multi-agent offline RL algorithm, which behaves as both a decentralized policy and a centralized controller including teammate modeling, and can be used for multi-agent trajectory prediction. Our experiments indicate strong performance compared with a set of recent offline MARL baselines on a variety of tasks.

Our proposed algorithm and experiments are limited in the following aspects: 1) MADIFF-D requires each agent to infer all other teammates' future trajectories, which is difficult and unnecessary in environments with a large number of agents. A potential modification is to infer a latent representation of others' trajectories; 2) We have not conducted experiments on datasets collected from environments with both high stochasticity and partial observability, such as SMACv2 (Ellis et al., 2022). In future work, we aim to address those limitations and further investigate conditioning on complex skills and constraints in multi-agent scenarios.

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

# Appendices

In this appendix, we give the pseudocode of multi-agent planning and multi-agent trajectory prediction with MADIFF model in Section A. In Section B, we demonstrate how multiple agents' trajectories are modeled by MADIFF during centralized control and decentralized execution in an example three-agent environment. In Section E, we provide details of the experiments, including the normalization used to compute the average score, the detailed network illustration unrolling each agent's U-Net, crucial hyperparameters, and an example of wall-clock time required for training MADIFF. In Section F, we demonstrate and analyze additional experimental results. Specifically, we provide ablation results to support the effectiveness of teammate modeling in MADIFF-D, show the quality of teammate modeling by MADIFF-D on SMAC tasks, and visualize predicted multi-player trajectories by MADIFF and the baseline algorithm on the NBA dataset.

## A  ALGORITHM

---

**Algorithm 1** Multi-Agent Planning with MADIFF

1: **Input:** Noise model $\epsilon_\theta$, inverse dynamics $I_\phi$, guidance scale $\omega$, history length $C$, condition $\boldsymbol{y}$
2: Initialize $h \leftarrow \texttt{Queue}(\texttt{length} = C)$; $t \leftarrow 0$           // Maintain a history of length C
3: **while** not done **do**
4:     Observe joint observation $\boldsymbol{o}$; $h.\texttt{insert}(\boldsymbol{o})$; Initialize $\boldsymbol{\tau}_K \sim \mathcal{N}(\mathbf{0}, \alpha\boldsymbol{I})$
5:     **for** $k = K \ldots 1$ **do**
6:        $\boldsymbol{\tau}_k[: \texttt{length}(h)] \leftarrow h$       // Constrain plan to be consistent with history
7:        **if** Centralized control **then**
8:           $\hat{\boldsymbol{\epsilon}} \leftarrow \epsilon_\theta(\boldsymbol{\tau}_k, k) + \omega(\epsilon_\theta(\boldsymbol{x}_k(\tau), \boldsymbol{y}, k) - \epsilon_\theta(\boldsymbol{\tau}_k, k))$       // Classifier-free guidance
9:           $(\boldsymbol{\mu}_{k-1}, \boldsymbol{\Sigma}_{k-1}) \leftarrow \texttt{Denoise}(\boldsymbol{\tau}_k, \hat{\boldsymbol{\epsilon}})$
10:       **else if** Decentralized execution **then**
11:         **for** agent $i \in \{1, 2, \ldots, N\}$ **do**
12:           $\hat{\epsilon}^i \leftarrow \epsilon_\theta^i(\tau_k^i, k) + \omega(\epsilon_\theta^i(\tau_k^i, y^i, k) - \epsilon_\theta^i(\tau_k^i, k))$       // Classifier-free guidance
13:           $(\mu_{k-1}^i, \Sigma_{k-1}^i) \leftarrow \texttt{Denoise}(\tau_k^i, \hat{\epsilon}^i)$
14:         **end for**
15:       **end if**
16:       $\boldsymbol{\tau}_{k-1} \sim \mathcal{N}(\boldsymbol{\mu}_{k-1}, \alpha\boldsymbol{\Sigma}_{k-1})$
17:     **end for**
18:     Extract $(\boldsymbol{o}_t, \boldsymbol{o}_{t+1})$ from $\boldsymbol{\tau}_0$
19:     **for** agent $i \in \{1, 2, \ldots, N\}$ **do**
20:        $a_t^i \leftarrow f_{\phi^i}(o_t^i, o_{t+1}^i)$
21:     **end for**
22:     Execute $\boldsymbol{a}_t$ in the environment; $t \leftarrow t + 1$
23: **end while**

---

**Algorithm 2** Multi-Agent Trajectory Prediction with MADIFF

1: **Input:** Noise model $\epsilon_\theta$, guidance scale $\omega$, condition $\boldsymbol{y}$, historical joint observations $h$ with length $C$, predict horizon $H$
2: Initialize $\boldsymbol{\tau}_K \sim \mathcal{N}(\mathbf{0}, \alpha\boldsymbol{I})$
3: **for** $k = K \ldots 1$ **do**
4:     $\boldsymbol{\tau}_k[: C] \leftarrow h$       // Constrain prediction to be consistent with history
5:     $\hat{\boldsymbol{\epsilon}} \leftarrow \epsilon_\theta(\boldsymbol{\tau}_k, k) + \omega(\epsilon_\theta(\boldsymbol{\tau}_k, \boldsymbol{y}, k) - \epsilon_\theta(\boldsymbol{\tau}_k, k))$       // Classifier-free guidance
6:     $(\boldsymbol{\mu}_{k-1}, \boldsymbol{\Sigma}_{k-1}) \leftarrow \texttt{Denoise}(\boldsymbol{\tau}_k, \hat{\boldsymbol{\epsilon}})$
7:     $\boldsymbol{\tau}_{k-1} \sim \mathcal{N}(\boldsymbol{\mu}_{k-1}, \alpha\boldsymbol{\Sigma}_{k-1})$
8: **end for**
9: Extract prediction $(\boldsymbol{o}_C, \boldsymbol{o}_{C+1}, \ldots, \boldsymbol{o}_{C+H-1})$ from $\boldsymbol{\tau}_0$

---

# B    ILLUSTRATION OF MULTI-AGENT TRAJECTORY MODELING

To provide a better understanding of how multiple agents' observations are modeled by MADIFF in centralized control and decentralized execution scenarios, we show illustrative examples in a typical three-agent environment in Fig. 4. If the environment allows for centralized control, we can condition MADIFF on all agents' historical and current observations, and let the model sample all agents' future trajectories as a single sample, as shown in Fig. 4(a). Then the current and next observations are sent to the inverse dynamics model for action prediction. If only decentralized execution is permitted, as shown in Fig. 4(b), agent 1 can only condition the model on its own information. The historical and current observations of other agents are masked when performing conditioning. MADIFF now not only generates agent 1's own future trajectories but also predicts the current and future observations of the other two agents. Due to the joint modeling of all agents during training, such predictions are also reasonable and can be considered as a form of teammate modeling from agent 1's perspective. Although teammate modeling is not directly used in generating agent 1's ego actions, it can help agent 1 refine its planned trajectories to be consistent with the predictions of others.

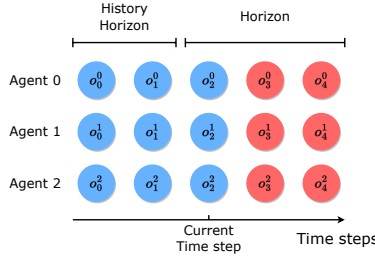

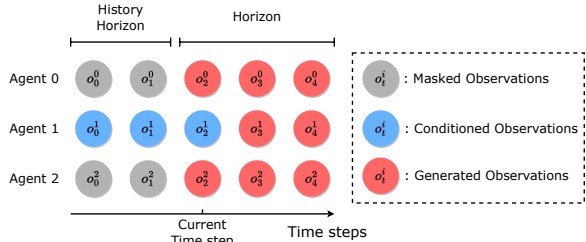

(a) MADIFF in centralized control.          (b) MADIFF in decentralized execution.

Figure 4: Illustration of how agents' observations are modelled by MADIFF in a three-agent environment. Note that figure (b) shows the situation when Agent 1 is taking action during decentralized execution.

# C    ADDITIONAL INFORMATION ON OFFLINE DATASETS

## C.1    MPE DATASETS

For MPE experiments, we use datasets and a fork of environment[1] provided by OMAR (Pan et al., 2022). They seem to be using an earlier version of MPE where agents can receive different rewards. For example, in the *Spread* task, team reward is defined using the distance of each landmark to its closest agent, which is the same for all agents. But when an agent collides with others, it will receive the team reward minus a penalty term. The collision reward has been brought into the team reward in the official repository since this commit[2]. However, the fork provided by OMAR still uses a legacy version. For fair and proper comparisons, we use OMAR's dataset and environment where all baseline models are trained and evaluated.

We have to note that different rewards for agents only happen at very few steps, which might not contradict the fully cooperative setting much. For example, OMAR's expert split of the *Spread* dataset consists of 1M steps, and different rewards are recorded only at less than 1.5% (14929) steps.

## C.2    SMAC DATASETS

For SMAC experiments, we adopt the off-the-grid dataset Formanek et al. (2023) and use *Good*, *Medium* and *Poor* datasets for each map. Each dataset is collected by three independently trained

---

[1] https://github.com/ling-pan/OMAR
[2] https://github.com/openai/multiagent-particle-envs/commit/
6ed7cac026f0eb345d4c20232bafa1dc951c68e7

QMIX policies to collect each dataset, and a small amount of exploration noise is added to the policies for enhanced behavioral diversity.

For visualizations of the distribution of episode returns in each dataset, we provide violin plots of all datasets we used in Fig. 5.

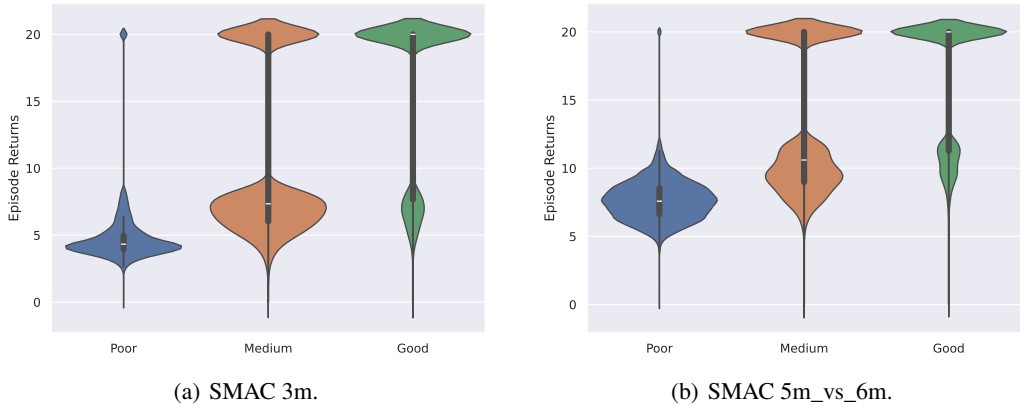

(a) SMAC 3m.  (b) SMAC 5m_vs_6m.

Figure 5: Violin plots of returns in SMAC datasets.

## D    BASELINE IMPLEMENTATIONS

In this section, we briefly describe how the baseline algorithms are implemented. For MATP experiments, we use the implementation from the official repository of Baller2Vec++[3]. Baseline results on MPE datasets are borrowed from Pan et al. (2022). According to their paper, they build all algorithms upon a modified version of MADDPG[4], which uses decentralized critics for all methods. Baselines on SMAC datasets are implemented by Formanek et al. (2023), and the performances are adopted from their reported benchmark results. The open-sourced implementation and hyperparameter settings can be found in the official repository[5].

## E    IMPLEMENTATION DETAILS

### E.1    SCORE NORMALIZATION

The average scores of MPE tasks in Tab. 1 are normalized by the expert and random scores on each task. Denote the original episodic return as $S$, then the normalized score $S_{\text{norm}}$ is computed as

$$S_{\text{norm}} = 100 \times (S - S_{\text{random}})/(S_{\text{expert}} - S_{\text{random}}) ,$$

which follows Pan et al. (2022) and Fu et al. (2020). The expert and random scores on Spread, Tag, and World are {516.8, 159.8}, {185.6, -4.1}, and {79.5, -6.8}, respectively.

### E.2    DETAILED NETWORK ARCHITECTURE

In Fig. 6, we unroll the U-Net structure of different agents in Fig. 1.

We describe the computation steps of attention among agents in formal. Each agent's local embedding $c^i$ is passed through the key, value, and query network to form $q^i$, $k^i$, and $v^i$, respectively. Then the dot product with scaling is performed between all agents' $q^i$ and $k^i$, which is followed by a Softmax operation to obtain the attention weight $\alpha^{ij}$. Each $\alpha^{ij}$ can be viewed as the importance of

---

[3]https://github.com/airalcorn2/baller2vecplusplus
[4]https://github.com/shariqiqbal2810/maddpg-pytorch
[5]https://github.com/instadeepai/og-marl

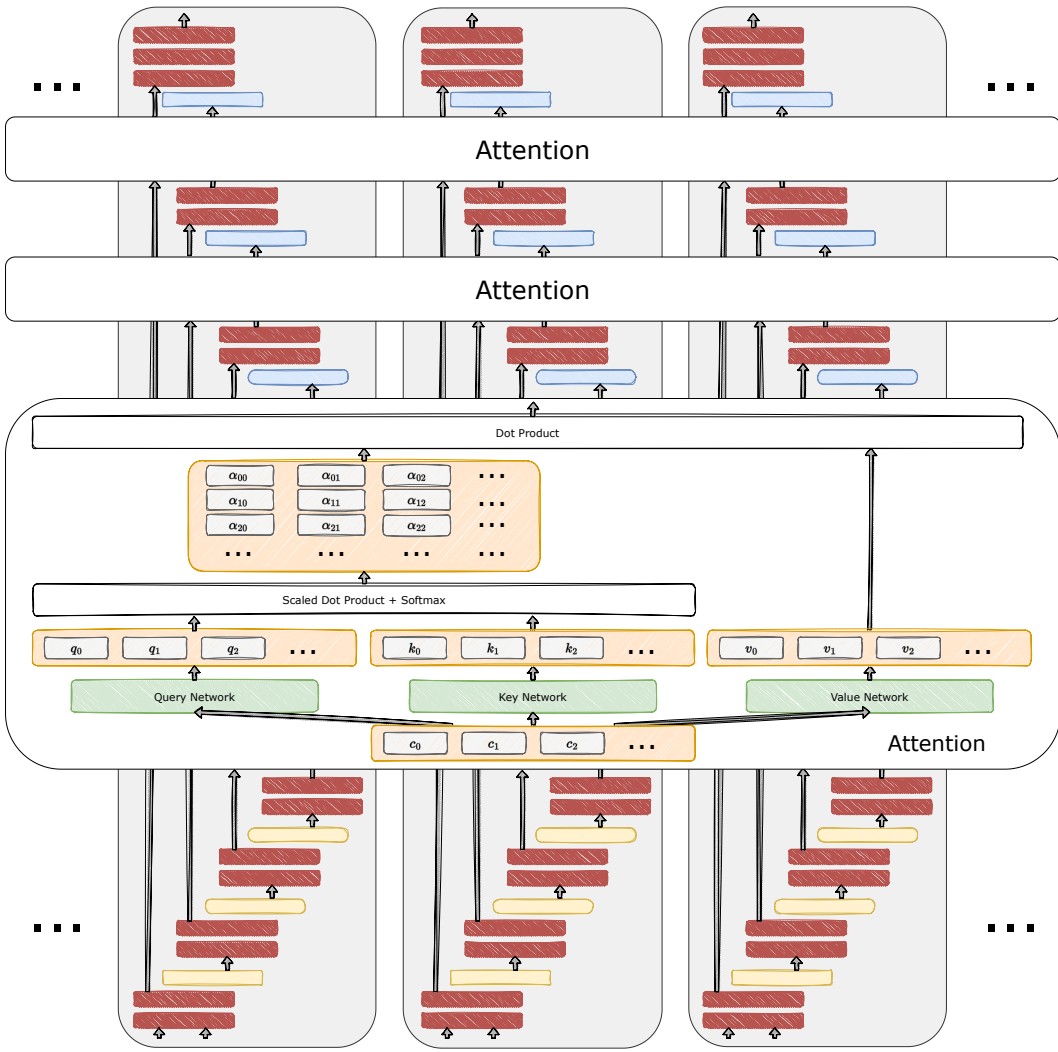

Figure 6: The detailed architecture of MADIFF. Each agent's U-Net is unrolled and lined up in the horizontal direction.

$j$-th agent to the $i$-th agent at the current time step. The second dot product is carried out between the weight matrix and the value embedding $v^i$ to get $\hat{c}^i$ after multi-agent feature interactions. Then $\hat{c}^i$ is skip-connected to the corresponding decoder block. The step-by-step computation of multi-agent attention in MADIFF can be written as

$$q^i = f_{\text{query}}(c^i),\ k^i = f_{\text{key}}(c^i),\ v^i = f_{\text{value}}(c^i)\ ;$$

$$\alpha^{ij} = \frac{\exp(q^i k^j / \sqrt{d_k})}{\sum_{p=1}^{N} \exp(q^i k^p / \sqrt{d_k})}\ ;$$

$$\hat{c}^i = \sum_{j=1}^{N} \alpha^{ij} v^j\ ,$$

where $d_k$ is the dimension of $k^i$.

### E.3 HYPERPARAMETERS

We list the key hyperparameters of MADIFF we used in Tab. 3 and Tab. 4. Return scale is the normalization factor used to divide the conditioned return before input to the diffusion model. The

rough range of the return scale can be determined by the return distributions of the training dataset. We only tune the guidance weight $\omega$, return scale, planning horizon $H$, and history horizon. We tried the guidance weight of $\{1.0, 1.2, 1.4, 1.6\}$, and found that different choices do not significantly affect final performances, in result we choose 1.2 as our default. We found larger planning horizon, in general, results in better performance. For MPE tasks, we find it unnecessary to condition on history observation sequence; thus, we set all history horizons to zero. In SMAC tasks which are more complex and more partially observable, we found a history horizon of 8, which is smaller than the planning horizon of 20, performs well across all datasets.

Table 3: Hyperparameters of MADIFF on MPE datasets.

| TestBed | Spread | | | | Tag | | | | World | | | |
|---|---|---|---|---|---|---|---|---|---|---|---|---|
| Dataset | Expert | Md-Replay | Medium | Random | Expert | Md-Replay | Medium | Random | Expert | Md-Replay | Medium | Random |
| Return scale | 700 | 500 | | | 700 | 600 | | 500 | 700 | 600 | | |
| Learning rate | 2e-4 | | | | | | | | | | | |
| Guidance scale $\omega$ | 1.2 | | | | | | | | | | | |
| Planning horizon $H$ | 24 | | | | | | | | | | | |
| History horizon | 0 | | | | | | | | | | | |
| Batch size | 32 | | | | | | | | | | | |
| Diffusion steps $K$ | 200 | | | | | | | | | | | |
| Optimizer | Adam Optimizer | | | | | | | | | | | |

Table 4: Hyperparameters of MADIFF on SMAC datasets.

| TestBed | 3m | | | 5m6m | | |
|---|---|---|---|---|---|---|
| Dataset | Good | Medium | Poor | Good | Medium | Poor |
| Return scale | 20 | | 10 | 20 | | 10 |
| Learning rate | 2e-4 | | | | | |
| Guidance scale $\omega$ | 1.2 | | | | | |
| Planning horizon $H$ | 20 | | | | | |
| History horizon | 8 | | | | | |
| Batch size | 32 | | | | | |
| Diffusion steps $K$ | 200 | | | | | |
| Optimizer | Adam Optimizer | | | | | |

### E.4 COMPUTING RESOURCES AND WALL TIME

We provide a concrete example as a reference for the time and resources required for training MADIFF. On a server with an AMD Ryzen 9 5900X (12 cores) CPU and an RTX 3090 GPU, we trained the MADIFF-C model on the Expert dataset from the MPE Spread task, which converged in about an hour. The curve of Wall-clock time spent for training and the corresponding model performance is shown in Fig. 7.

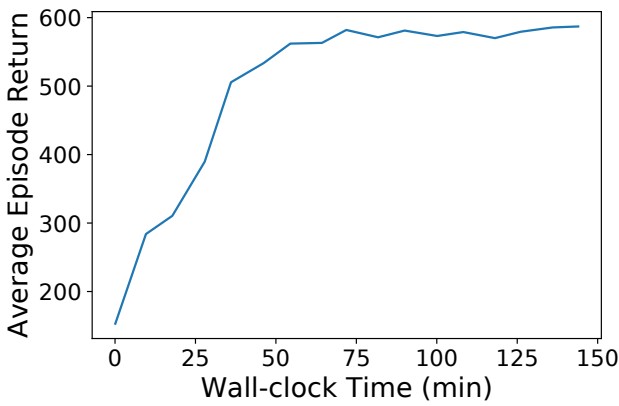

Figure 7: Wall-clock time and corresponding average episode return (average over 10 episodes) during training MADIFF-C for MPE Spread task.

## F ADDITIONAL EXPERIMENTAL RESULTS

### F.1 EFFECTIVENESS OF TEAMMATE MODELING

To investigate whether teammate modeling can lead to performance improvements during decentralized execution, we conduct ablation experiments on MPE Spread datasets. We compare MADIFF-D with its variant that adopts the same network architecture but masks the diffusion loss on other agents' trajectories during training. We denote the variant as MADIFF-D w/o OM. The results are presented in Tab. 5, which show that teammate modeling results in notable performance improvements on all four levels of datasets.

Table 5: Ablation results of teammate modeling on MPE Spread datasets across 3 seeds.

| Dataset | MADIFF-D w/o OM | MADIFF-D |
|---------|-----------------|----------|
| Expert | $93.4 \pm 3.6$ | $\mathbf{98.4 \pm 12.7}$ |
| Medium | $35.4 \pm 6.6$ | $\mathbf{53.2 \pm 2.3}$ |
| Md-Replay | $17.7 \pm 4.3$ | $\mathbf{42.9 \pm 11.6}$ |
| Random | $5.7 \pm 3.1$ | $\mathbf{19.4 \pm 2.9}$ |

### F.2 TEAMMATE MODELING ON SMAC TASKS

We show and analyze the quality of teammate modeling by MADIFF-D on SMAC. Specifically, we choose two time steps from an episode on 3m map to analyze predictions on allies' attack targets and health points (HP), respectively.

On top of Fig. 8(a) is attacked enemy agent ID (0, 1, 2 stands for E0, E1, E2) of ally agents A0, A1, and A2. The first row is the ground-truth ID, and the second and the third rows are the predictions made by MADIFF-D from the other two allies' views. We can see that the predictions are in general consistent with the ground-truth ID. As can be seen from the true values of the attack enemy ID, agents tend to focus their firepower on the same enemies at the same time. And the accurate prediction of allies' attack enemy IDs intuitively can help to execute such a strategy.

In Fig. 8(b) we visualize the HP change curve of ally agents starting from another time step. From the environment state visualization below, agent A2 is the closest to enemies, so its HP drops the fastest. Such a pattern is successfully predicted by the other two agents.

### F.3 PREDICTED TRAJECTORY VISUALIZATION ON NBA DATASET

We visualize the players' moving trajectories predicted by MADIFF-C and Baller2Vec++ on the NBA dataset in Fig. 9. In each image, the solid lines are real trajectories and the dashed lines are trajectories predicted by the model. The trajectories predicted by MADiff-C are closer to the real trajectories and are overall smoother compared to the Baller2Vec++ predictions.

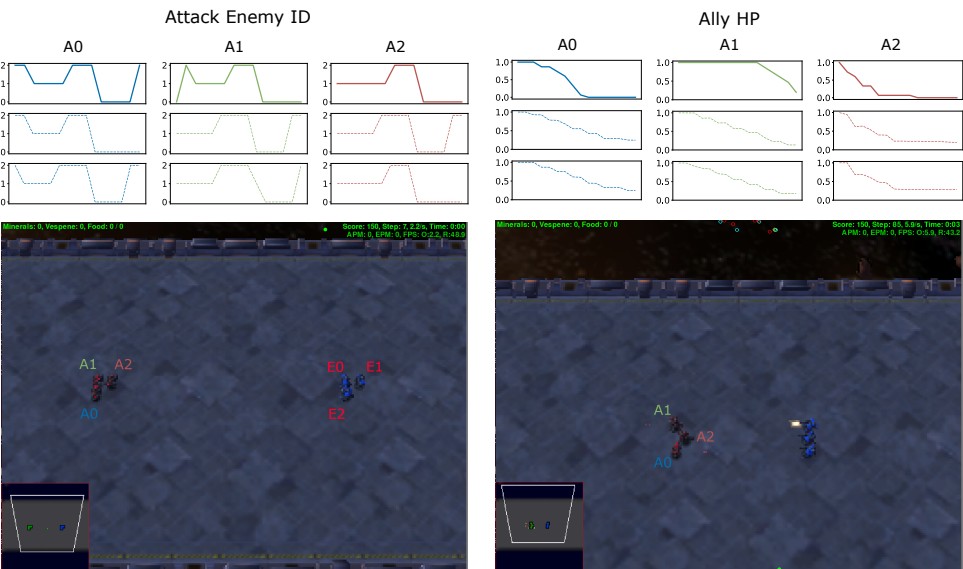

(a) Ground-truth and predicted enemy's ID to attack by each ally agent.

(b) Ground-truth and predicted health points (HP) of each ally agent.

Figure 8: The ground-truth and predicted information of different MADIFF agents at two-time step. On the top of each figure, each column describes a different agent. The first row shows the change curve of the real value, and the last two rows below are the information predicted by other agents.

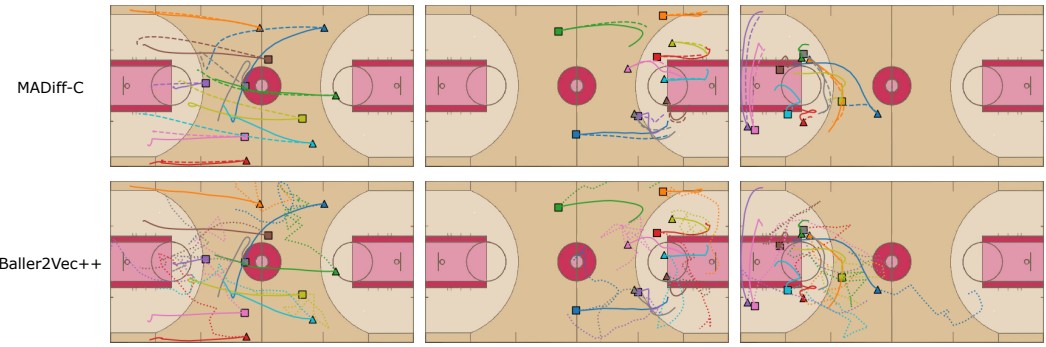

Figure 9: Real and Predicted multi-player trajectories by MADIFF-C and Baller2Vec++.

