# OpenReview forum: "MADiff: Offline Multi-agent Learning with Diffusion Models"
_ICLR.cc/2024/Conference — Submitted to ICLR 2024_

### Official Review · Reviewer_Tx3W · 2023-11-01

**Soundness:** 2 fair
**Presentation:** 2 fair
**Contribution:** 3 good
**Rating:** 5
**Confidence:** 4

**Summary:**

The paper considers cooperative multi-agent problems within the centralized training decentralized execution paradigm. The paper proposes the first multi-agent method to use a diffusion model in order to generate trajectories during online execution. The advantage of using a generative model to create policies is that the diffusion method is able to learn from behavioral data with severe extrapolation errors. Prior value-based methods are unable to do this due to the updates from out-of-distribution points. Additionally, prior methods have yet to apply this type of method to multi-agent cases. The paper introduces a novel attention-based diffusion model for multi-agent learning which combines CTDE, opponent modeling, and trajectory prediction. The paper evaluates on benchmark CTDE environments with offline data available and finds that their method tends to perform better.

**Strengths:**

+ Using diffusion to model the agents in multi-agent learning is a very interesting premise. There is great potential for this method to be quite impactful in a multi-agent learning context. The ability to generalize from the value-based methods may prove useful given a way to combine this with online finetuning. This may be interesting for future work on foundation models in this area.

**Weaknesses:**

- This is a very interesting paper but there are unfortunate issues with the experimental design. SMAC (Version 1) has known issues. There is a recent version that was recently accepted at NeurIPS but has been released for a while that fixes the “open-loop” issues of SMACv1 called SMACv2[1]. Given that these tasks were essentially solved by the community with in the easy maps that are compared with in the results, the results may not be as convincing as currently indicated. Why is the score for SMAC so low. Typically the SMAC reported value is winrate, which should be near 100% for these easier SMAC tasks, especially with QMIX. There must be a bug in the evaluation or the data is far to limited to evaluate the method properly.
[1] Ellis, B., Moalla, S., Samvelyan, M., Sun, M., Mahajan, A., Foerster, J. N., & Whiteson, S. (2022). SMACv2: An improved benchmark for cooperative multi-agent reinforcement learning. arXiv preprint arXiv:2212.07489.

- Regarding the other experimental design items, it is unclear what hypothesis the MPE environments test that would not be tested by SMAC. For the MATP, this seems like a niche task. It is unclear if the test games have a distribution of trajectories that are out of distribution of the training/validation set. Otherwise, it is hard to understand the generalization capability of the method.

- How does the centralized control version scale with the number of agents? It is unclear if this is a useful avenue of exploration compared with the CTDE paradigm.

**Questions:**

I still do not understand the “Decentralized policy and centralized controller” terminology. I assumed that this means centralized training decentralized execution (CTDE). The new language is confusing.

Is an attention network used for all comparison baselines?

Please see Weaknesses for additional questions.

---

> ### Author Response · Authors · 2023-11-21
>
> Dear reviewer, thank you very much for your constructive comments! We try to address your concerns as follows.
>
> **Q1: "Given that these tasks were essentially solved by the community with in the easy maps that are compared with in the results, the results may not be as convincing as currently indicated."**
>
> **A1:**
> Thanks for your comment, but you might misunderstand our experimental settings. Our method and all baselines, including QMIX, are trained in a fully offline manner. Offline Policy learning is far more challenging than online learning due to limited data support and sub-optimal behavior policies used to collect the dataset.
>
> One task can be solved by online RL methods does not mean it can be solved at the expert level with offline learning. For example, PPO [1], which was proposed in 2017, can easily achieve expert-level performances in single-agent mujoco environments (Hopper, HalfCheetah, ...). However, the widely-used D4RL benchmark [2], which was released in 2020, still uses those environments to build their dataset. And the SOTA algorithms today can only obtain about half of expert scores on D4RL's halfCheetah-medium dataset [3].
>
> As a result, our results under offline MARL settings should not be compared with online MARL methods.
>
> **Q2: "Regarding the other experimental design items, it is unclear what hypothesis the MPE environments test that would not be tested by SMAC."**
>
> **A2:**
> Both MPE and SMAC datasets are widely adopted by existing works in offline MARL [4, 5]. These two classes of environments have distinct state/action spaces and require different cooperation patterns. We include both of them to ensure a thorough evaluation of our method. Two intuitive differences are that the action space of MPE is continuous, and the initial state distributions are more diverse than SMAC.
>
> **Q3: "It is unclear if the test games have a distribution of trajectories that are out of distribution of the training/validation set."**
>
> **A3:**
> Thanks for your good question! We adopt the standard training/validation/test split in MATP experiments, which allows for out-of-distribution (OOD) tests. On offline MARL tasks, our evaluation procedure follows the existing works on offline RL [6, 7] and MARL [4, 5] and does not deliberately test the generalization ability.
>
> We want to highlight that different from static supervised learning tasks in MATP, evaluating offline-learned policies by interacting online naturally includes the requirement of OOD generalizations. Due to errors induced by functional approximations and stochasticity in environmental dynamics, agents will encounter OOD states even if they are started from in-distribution states. The learned policies must have the ability to recover from OOD states and prevent errors from accumulating over time.
>
> Designing reasonable benchmarks for specifically evaluating the generalization ability of offline MARL models is less discussed yet, and we acknowledge that it is an important future research direction.
>
> **Q4: "How does the centralized control version scale with the number of agents? It is unclear if this is a useful avenue of exploration compared with the CTDE paradigm."**
>
> **A4:**
> Both versions of MADiff have their own values. MADiff-C is only applicable in cases where a centralized controller for all agents is allowed, such as traffic light control or fleet control. Since all agents' actions are made jointly, it is easier to keep their behaviors coordinated. Therefore, MADiff-C is supposed to perform better compared to MADiff-D in those cases, which is aligned with our experiment results. However, there are more scenarios where centralized controls are impossible, e.g., autonomous driving and distributed sensor networks [8]. MADiff-D, which is the CTDE version of MADiff, is designed for these scenarios.
>
> MADiff-C and MADiff-D scale the same as the number of agents increases. We use a shared U-Net model for all agents, where different agents' data can be batched together and passed through the network. It does not cost much more inference time as the number of agents increases with GPU-accelerated computing. For self-attention among agents' embeddings, it can also scale with linear complexity [9].
>
> When the number of agents is extremely large, we can modify the opponent/teammate modeling to operate in a projected low-dimensional space with a latent diffusion model [10]. We leave this direction to future work.

---

> > ### Author Response · Authors · 2023-11-21
> >
> > **Q5: "I still do not understand the “Decentralized policy and centralized controller” terminology. I assumed that this means centralized training decentralized execution (CTDE). The new language is confusing."**
> >
> > **A5:**
> > As we mentioned in **A4**, decentralized policy and centralized controller refer to different control styles, which correspond to MADiff-D and MADiff-C, respectively. MADiff-D can be classified as a CTDE method, where the diffusion model is trained on all agents' data and used by each agent independently during execution.
> >
> > The terminology of "decentralized policy" and "centralized controller" is not novel in the domain of MAL [11, 12]. We confirm that our usage of these terms is consistent with existing literature.
> >
> > **Q6: "Is an attention network used for all comparison baselines?"**
> >
> > **A6:**
> > Thanks for the question. In general, the network design of baseline algorithms follows their original paper, and we have included a detailed description of baseline implementations in the Appendix.
> >
> > **References**
> >
> > [1] Schulman, John, et al. "Proximal policy optimization algorithms." *arXiv preprint arXiv:1707.06347* (2017).
> >
> > [2] Fu, Justin, et al. "D4rl: Datasets for deep data-driven reinforcement learning." *arXiv preprint arXiv:2004.07219* (2020).
> >
> > [3] Ajay, Anurag, et al. "Is Conditional Generative Modeling all you need for Decision Making?." *The Eleventh International Conference on Learning Representations*. 2022.
> >
> > [4] Pan, Ling, et al. "Plan better amid conservatism: Offline multi-agent reinforcement learning with actor rectification." *International Conference on Machine Learning*. PMLR, 2022.
> >
> > [5] Tian, Qi, et al. "Learning from good trajectories in offline multi-agent reinforcement learning." *Proceedings of the AAAI Conference on Artificial Intelligence*. Vol. 37. No. 10. 2023.
> >
> > [6] Kumar, Aviral, et al. "Conservative q-learning for offline reinforcement learning." *Advances in Neural Information Processing Systems* 33 (2020): 1179-1191.
> >
> > [7] Fujimoto, Scott, and Shixiang Shane Gu. "A minimalist approach to offline reinforcement learning." *Advances in neural information processing systems* 34 (2021): 20132-20145.
> >
> > [8] Olfati-Saber, Reza, and Jeff S. Shamma. "Consensus filters for sensor networks and distributed sensor fusion." *Proceedings of the 44th IEEE Conference on Decision and Control*. IEEE, 2005.
> >
> > [9] Shen, Zhuoran, et al. "Efficient attention: Attention with linear complexities." *Proceedings of the IEEE/CVF winter conference on applications of computer vision*. 2021.
> >
> > [10] Rombach, Robin, et al. "High-resolution image synthesis with latent diffusion models." *Proceedings of the IEEE/CVF conference on computer vision and pattern recognition*. 2022.
> >
> > [11] Zhang, Kaiqing, et al. "Fully decentralized multi-agent reinforcement learning with networked agents." *International Conference on Machine Learning*. PMLR, 2018.
> >
> > [12] Lin, Alex Tong, et al. "Decentralized multi-agents by imitation of a centralized controller." *Mathematical and Scientific Machine Learning*. PMLR, 2022.

---

> > > ### Comment · Reviewer_Tx3W · 2023-11-21
> > >
> > > Thank you for the further responses.
> > >
> > > Regarding Q6, I do not believe that this is a fair comparison with respect to empirical comparison. It is hard to say whether the diffusion aspect or the attention aspect is helping the method increase the performance. If a comparison of just your method with and without the attention mechanism existed, would there be an increase in performance? I think the use of diffusion is very novel and interesting, but I am having trouble being convinced that the results are strictly due to the diffusion aspect.

---

> > > > ### Author Response · Authors · 2023-11-22
> > > > **Responses to remaining concerns**
> > > >
> > > > Thank you very much for the prompt replies, and we sincerely appreciate your re-rating of our work. We address your remaining concerns in the following responses.
> > > >
> > > > **Q1: The [1] benchmark that provides the data also has an SMACv2 available ... Have you tested your algorithm with that method?**
> > > >
> > > > **A1:** Thanks for your comment! We totally agree that SMACv2 is an important benchmark, and we are running experiments on datasets of zerg_5_vs_5 and terran_10_vs_10 from SMACv2 and will update the results to the manuscript after they are available.
> > > >
> > > > May we explain the reasons of why we chose SMACv1 rather than SMACv2 in the first place:
> > > >
> > > > + Prior works in offline MARL reported their results on maps from the SMACv1 environment rather than v2;
> > > >
> > > > + In the released off-the-grid dataset, each SMACv2 map only comes with one *Replay* dataset, whereas each v1 map has three available datasets: *Good*, *Medium*, and *Poor*.
> > > >
> > > > On the other hand, we would like to discuss the impacts of the stochasticity from environments on our algorithm:
> > > >
> > > > + The initial states in MPE tasks are randomly sampled, and MADiff clearly surpasses baseline algorithms on those tasks;
> > > >
> > > > + Diffusion models are pretty good at modeling complex distributions, and intuitively, MADiff will not suffer much from the stochasticity of SMACv2 compared to the baseline algorithm.
> > > >
> > > > **Q2: "Please indicate the meaning of the score. Do the offline methods learn from adequately performing data? Based on [1], the relationship between winrate and score is unclear."**
> > > >
> > > > **A2:** We apologize for not fully understanding your question in prior responses.
> > > >
> > > > The scores we reported are the undiscounted summations of per-step team rewards in an episode. According to the original paper introducing the SMAC benchmark [1], per-step rewards are computed as the hit-point damage dealt and bonuses of 10 and 200 points for killing each enemy unit and winning the game, respectively. The rewards are scaled so that the maximum cumulative reward (score) achievable in each scenario is around 20. We checked the SMAC environment code (https://github.com/oxwhirl/smac) used in our experiments and confirmed that it is consistent with the above description.
> > > >
> > > > From the magnitude of the different reward terms, it can be seen that winning or not significantly impacts score calculations. Compared to win rate, average scores can distinguish between failed episodes, since the number of enemies killed and hit-point damage dealt are considered.
> > > >
> > > > As can be seen from the added violin plots in Appendix C, datasets in the same map have different return/score distributions. *Good* datasets consist of a majority of expert-level trajectories with scores around the maximum score of 20. Only a few trajectories in *Meidum* datasets are optimal, and *Poor* datasets do not include expert trajectories.
> > > >
> > > > **Q3: "Regarding Q6, I do not believe that this is a fair comparison with respect to empirical comparison. It is hard to say whether the diffusion aspect or the attention aspect is helping the method increase the performance ... but I am having trouble being convinced that the results are strictly due to the diffusion aspect."**
> > > >
> > > > **A3:**
> > > >
> > > > Thank you for the questions.
> > > >
> > > > First, we need to clarify that both diffusion modeling and attention networks are indispensable components of our proposed algorithm. We will highlight this point in the revised manuscript based on our ablation results.
> > > >
> > > > In our reply to **Reviewer Khwp**'s Q5, we compared the performances of MADiff and ConcatDiff in the CTCE setting. ConcatDiff is an ablation variant that does not use attention networks but directly concatenates all agents' observations as input to a U-Net. Results show that MADiff-C clearly outperforms ConcatDiff, suggesting that **directly migrating a single-agent diffusion network structure to a multi-agent learning problem will not yield good results**. To validate whether this argument applies in the CTDE setting, we are running experiments with ConcatDiff-D where other agents' observations are masked before concatenation during training, and we will soon post the results of its comparison with MADiff-D.
> > > >
> > > > Baseline algorithms based on multi-agent Q learning typically use single-agent independent actors, and only their centralized critics can incorporate attention modules. A strictly aligned comparison of network designs is impossible since critics are only used during training, and our diffusion network is used for training and online rollouts. Besides, to the best of our knowledge, **no existing work uses attention-based critics in the context of offline MARL**. Besides Q learning methods, MADT views MAL problems as sequence modeling and uses a fully attention-based network. In our added experiments, MADiff performed better than MADT in most datasets.
> > > >
> > > >
> > > >
> > > > **References**
> > > >
> > > > [1] Samvelyan, Mikayel, et al. "The StarCraft Multi-Agent Challenge." *Proceedings of the 18th International Conference on Autonomous Agents and MultiAgent Systems*. 2019.

---

> > ### Comment · Reviewer_Tx3W · 2023-11-21
> >
> > Thank you for the reply. Given that some concerns (but not all) have been addressed. I have readjusted my score.
> >
> > Q1: "Given that these tasks were essentially solved by the community within the easy maps that are compared with in the results, the results may not be as convincing as currently indicated."
> >
> > To clarify, the question asked was the following: "Why is the score for SMAC so low? Typically the SMAC reported value is winrate,"
> >
> > Please indicate the meaning of the score. Do the offline methods learn from adequately performing data? Based on [1], the relationship between winrate and score is unclear.
> >
> > The [1] benchmark that provides the data also has an SMACv2 available. Since SMACv2 requires a closed-loop policy, it is much more interesting for comparison even in the offline reinforcement learning case. Additionally, there is data readily available. Have you tested your algorithm with that method?
> >
> > [1] Claude Formanek, Asad Jeewa, Jonathan Shock, and Arnu Pretorius. Off-the-grid marl: a framework
> > for dataset generation with baselines for cooperative offline multi-agent reinforcement learning.
> > arXiv preprint arXiv:2302.00521, 2023.

---

> ### Author Response · Authors · 2023-11-23
> **Updates of ablation results**
>
> We are delighted to share with you some new ablation results!
>
> As mentioned in **A3**, to test whether both diffusion modeling and attention networks are important in the success of MADiff, we compare our network design to ConcatDiff, which is an ablation variant that does not use attention networks but directly concatenates all agents' observations as inputs to a U-Net. ConcatDiff-C is used for centralized control, and ConcatDiff-D masks other agents' observations before concatenation during training. Results are presented as follows:
>
> | Dataset          | OMAR            | ConcatDiff-D    | MADiff-D        | ConcatDiff-C | MADiff-C    |
> | :--------------- | :-------------- | :-------------- | --------------- | ------------ | ----------- |
> | Spread-Expert    | 114.9 $\pm$ 2.6 | 106.4 $\pm$ 3.5 | 98.4 $\pm$ 12.7 | 112.7 $\pm$ 4.3  | 114.7 $\pm$ 5.3 |
> | Spread-Md-Replay | 37.9 $\pm$ 12.3 | 31.5 $\pm$ 2.7  | 42.9 $\pm$ 11.6 | 32.2 $\pm$ 3.5   | 47.2 $\pm$ 6.6  |
>
> We can observe that ConcatDiff performs worse than MADiff in both CTCE and CTDE cases, and underperforms OMAR, which is the best baseline method in MPE datasets.
>
> We sincerely hope that our responses and the additional experiments we conducted have addressed your concerns and enhanced the quality of the paper. We would greatly appreciate it if you could clarify any remaining concerns you may have.

---

### Official Review · Reviewer_QCKf · 2023-11-01

**Soundness:** 2 fair
**Presentation:** 2 fair
**Contribution:** 2 fair
**Rating:** 5
**Confidence:** 4

**Summary:**

This paper proposes MADiff which incorporates diffusion models for either a centralized planner or decentralized actors. The diffusion model builds on Decision Diffuser (Ajay et al., 2023) and additionally uses an attention layer for better coordination among the agents. The training procedure enables the decentralized policies to have opponent modeling capabilities where agents predict the observations of other agents. MADiff is evaluated on SMAC and MPE as well as the NBA Dataset for trajectory prediction.

**Strengths:**

The use of diffusion models for offline MARL is interesting and this combination is, as far as I know, novel. Diffusion models have shown promise in offline RL due to its potential for stitching and generalization capabilities. In this context, whether diffusion policies can improve offline MARL is an important problem, and it is interesting to view this from the lens of opponent modeling. Furthermore, viewing offline MARL as trajectory prediction makes it more scalable without requiring restrictive assumptions such as IGM. Finally, the attention mechanism which is the main difference between Decision Diffuser and MADiff seems important in the offline MARL setting, as shown in the ablation studies (Section 5.5).

**Weaknesses:**

1. The overall structure of the paper and algorithm is very similar to Decision Diffuser (Ajay et al. 2023), but with an additional attention layer.
2. The benefits of using diffusion policies mentioned in Decision Diffuser, such as generating novel behaviors, stitching or different conditions for $y(\tau)$ such as constraints or skills are not addressed in MADiff in the context of MARL.


3. The inverse dynamics loss in Eq. 7 looks a little strange since the observations $s^i, s^{i’}$ are used, which makes it unclear what this is predicting, since other agents’ actions as well as partial observations are involved. This is essentially assuming that the next state observation only conditions on the current local observations and local actions. This may be fine for simple tasks but may not translate to more complex tasks where the other agents’ actions impact the dynamics. Also, the notation $s^i$ is confusing here since Section 2 uses $o^i$.
4. Only a small number of agents (3-5) are considered in the experiments .
5. While opponent modeling is mentioned as one of the contributions, previous work in opponent modeling are missing from the Related Work section (e.g. [1] and references therein) and there is no explicit mention about what kinds of information are available at test time (e.g. only local observation-action history of ego-agent).
6. OMAR is excluded from SMAC tasks as a baseline despite OMAR also reporting SMAC results.
7. (Minor) The perturbed states $\tilde x$ in $\hat \tau_k := [s_t,\tilde x_{t},... ]$ above Eq. 4 are not properly defined.
8. (Minor) Grammatical error in 5.4: “We that..”

[1] Agent Modelling under Partial Observability for Deep Reinforcement Learning (Papoudakis et al,. NeurIPS 2021)

**Questions:**

1. How do diffusion models address the curse of dimensionality of the joint action space? How does MADiff scale as $N$ increases?
2. In the second paragraph of the Introduction, do you mean that “learning the $\textbf{incorrect}$ centralized value for each agent…”? Does this mean that extrapolation error can occur even when the correct value is learned, or is this a typo?
3. Please address the points raised in Weaknesses.

---

> ### Author Response · Authors · 2023-11-21
>
> Dear reviewer, we appreciate your time and effort in reviewing this paper. We try to address your concerns as follows and hope we can ease your concerns.
>
> **Q1: "The overall structure of the paper and algorithm is very similar to Decision Diffuser, but with an additional attention layer."**
>
> **A1:**
> It is indeed that many successful design choices of [1] have inspired us and guided us to derive our method. Similarly, much of [1] also referred to the design of diffuser [2], but that doesn't take away from the fact that it's an excellent research.
>
> Compared with [1], MADiff, which is the first to apply diffusion models in offline MARL, is naturally designed for multi-agent problems, taking multi-agent coordination, opponent modeling, and trajectory planning in a unified framework. **Our contributions include but beyond the model design**, and the most important one is that the proposed framework allows the use of the same model structure to handle various tasks (CTCE, CTDE, MATP) by flexible conditioning during evaluation and achieves superior results on those tasks.
>
> **Q2: "The benefits of using diffusion policies mentioned in Decision Diffuser, ... are not addressed in MADiff in the context of MARL."**
>
> **A2:**
> Compared to single-agent learning, multi-agent learning problems themselves are more challenging and can benefit from diffusion modeling. Specifically, besides the variety of a single agent's behavior, the interplay between multiple agents exhibits a multi-modal and complex nature. Our goal is to introduce DM with powerful modeling ability of arbitrary distributions to MAL for better learning multi-agent interactions in offline datasets. Verifying whether skill composition in SAL still applies in MAL settings is an interesting trial, but it is out of the scope of our paper.
>
> **Q3: "The inverse dynamics loss in Eq. 7 looks a little strange since the observations s^i, s^{i'} are used, ... but may not translate to more complex tasks where the other agents’ actions impact the dynamics. Also, the notation s^i is confusing here since Section 2 uses o^i."**
>
> **A3:**
> Thanks for your careful checking. There is a typo in Eq.7, where $s^i$ and $s^{i'}$ should be replaced by $o^i$ and $o^{i'}$. We have made corrections in the revised manuscript.
>
> We would like to justify our use of IDM $I(o^i, o^{i'})$ in multi-agent tasks in two aspects:
>
> + First, since the inverse dynamics model is used rather than a forward dynamics model, we are not assuming "the next state observation only conditions on the current local observations and local actions", but that the current and next local observations determine the local actions. Since the next local observation is jointly generated with predictions of other agents' observations, it already considers the impact of other agents' actions.
>
> + More importantly, the goal in Dec-POMDP is learning a policy for each agent, which can maximize cumulative rewards **when all agents are controlled by learned policies** [3]. Therefore, although actions are made in a decentralized way, each agent's policy is learned from the same dataset the IDM is trained on. Thus, $p(a^i|o^i, o^{i'})$ during evaluation should be similar to IDM's training distribution. Learning policies that can coordinate with any teammates requires zero-shot coordination [4], and is (to our knowledge) not yet discussed in offline settings and clearly out of the scope of this paper.
>
> **Q4: "Only a small number of agents (3-5) are considered in the experiments."**
>
> **A4:**
> Most existing works in offline MARL do not consider tasks with a large number of agents. For example, MADT-KD [5] and OMAR [6] are tested on environments with up to 8 agents. Moreover, OMAR uses eight multi-agent environments for benchmarking, yet only two of these consist of more than 5 agents. Despite the fact that we consider at most 5 agents in MARL experiments, our MATP experiments are conducted on NBA datasets with 10 agents, and MADiff demonstrates superior performance than the baseline method.
>
> **Q5: "While opponent modeling is mentioned as one of the contributions, previous works in opponent modeling are missing from the Related Work section."**
>
> **A5:**
> Thanks for your suggestions! We have included a discussion of previous opponent modeling literature in the related work section.

---

> > ### Author Response · Authors · 2023-11-21
> >
> > **Q6: "there is no explicit mention about what kinds of information are available at test time"**
> >
> > **A6:**
> > For the centralized control setting, a centralized model has access to all agents' local observations. When only decentralized execution is allowed, each agent makes decisions based on its own local observations, which is mentioned in the "decentralized execution" paragraph in Section 3.3. In some cases (e.g., SMAC), models are conditioned on local observation history instead of only current local observation, as we mentioned in the last paragraph of Section 3.3. We have revised the first paragraph in Section 3.3 to more clearly state the information available in the centralized control setting.
> >
> > **Q7: "OMAR is excluded from SMAC tasks as a baseline despite OMAR also reporting SMAC results."**
> >
> > **A7:**
> > Although the OMAR paper reports results on SMAC, the authors' implementation of OMAR (https://github.com/ling-pan/OMAR) only contains the algorithm on continuous-action environments. Also, they have not released the dataset they used for SMAC experiments. Since OMAR relies on zeroth-order optimizations on the action space, the modifications required for handling discrete actions are non-trivial.
> >
> > We tried our best to reimplement a discrete version of OMAR, but the results on our SMAC datasets are much worse than other baseline methods. Therefore, we decided to exclude OMAR from SMAC datasets. It is worth noting that MA-ICQ and MA-CQL are both strong baselines, and we believe the results are sufficient to demonstrate the competitive performances of our method.
> >
> > **Q8: Minor issues.**
> >
> > **A8:**
> > Thanks for your careful checking! We have fixed the grammar error and revised the notation of denoised samples.
> >
> > **Q9: "How do diffusion models address the curse of dimensionality of the joint action space? How does MADiff scale as N increases?"**
> >
> > **A9:**
> > For all our experiments, we use a shared U-Net model for all agents, where different agents' data can be batched together and passed through the network. It does not cost much more inference time as the number of agents increases with GPU-accelerated computing. For self-attention among agents' embeddings, it can also scale with linear complexity [5].
> >
> > When the number of agents is extremely large, we can modify the opponent/teammate modeling to operate in a projected low-dimensional space with a latent diffusion model [6], thus avoiding the curse of dimensionality. We leave this direction to future work.
> >
> > **Q10: "In the second paragraph of the Introduction, do you mean that 'learning the incorrect centralized value for each agent…'? Does this mean that extrapolation error can occur even when the correct value is learned, or is this a typo?"**
> >
> > **A10:**
> > This is a typo and we apologize for the mistake. It has been fixed in the revised manuscript.
> >
> >
> > **References**
> >
> > [1] Ajay, Anurag, et al. "Is Conditional Generative Modeling all you need for Decision Making?." *The Eleventh International Conference on Learning Representations*. 2022.
> >
> > [2] Janner, Michael, et al. "Planning with Diffusion for Flexible Behavior Synthesis." *International Conference on Machine Learning*. PMLR, 2022.
> >
> > [3] Rashid, Tabish, et al. "Monotonic value function factorisation for deep multi-agent reinforcement learning." *The Journal of Machine Learning Research* 21.1 (2020): 7234-7284.
> >
> > [4] Treutlein, Johannes, et al. "A new formalism, method and open issues for zero-shot coordination." *International Conference on Machine Learning*. PMLR, 2021.
> >
> > [5] Shen, Zhuoran, et al. "Efficient attention: Attention with linear complexities." *Proceedings of the IEEE/CVF winter conference on applications of computer vision*. 2021.
> >
> > [6] Rombach, Robin, et al. "High-resolution image synthesis with latent diffusion models." *Proceedings of the IEEE/CVF conference on computer vision and pattern recognition*. 2022.

---

> ### Author Response · Authors · 2023-11-23
>
> Dear Reviewer QCKf,
>
> We deeply appreciate the valuable and constructive feedback you provided. As the rebuttal discussion between reviewers and authors is coming to a close on November 23rd, we would greatly appreciate it if you could clarify any remaining concerns you may have. This will ensure that we can adequately address them during this period.
>
> Thank you for your attention to this matter. We are looking forward to your response.
>
> Best regards,
>
> The Authors

---

### Official Review · Reviewer_nDb9 · 2023-11-02

**Soundness:** 3 good
**Presentation:** 4 excellent
**Contribution:** 3 good
**Rating:** 6
**Confidence:** 3

**Summary:**

This paper introduces a novel diffusion-based offline multi-agent learning framework, called MADIFF. Specifically, MADIFF proposes the attention-based architecture in Section 3.1 to interchange information and learn coordination between agents. MADIFF is also designed with centralized training and decentralized execution, inherently enabling the framework to perform the opponent modeling during execution. Evaluations in MPE, SMAC, and MATP show the effectiveness of MADIFF against competitive baselines.

**Strengths:**

1. Overall, the paper is very clearly written achieving the SOTA results compared to baselines.
2. MADIFF is a principled diffusion-based framework without needing complicated components to achieve effective performance.
3. Code is available for reproducibility.

**Weaknesses:**

Overall, I enjoyed reading this paper and learning about MADIFF, but there are a few questions that I would like to follow up on:
1. Because MADIFF needs to predict the next observations (Equations 8 and 9), I would like to ask whether MADIFF can be applied to domains with high-dimensional inputs (e.g., image). I noticed that the experimental domains have relatively small dimension inputs.
2. In Equation 9, inferring all other agents' observations could be very difficult when $N$ is large. Could MADIFF scalability benefit by inferring only the agent $i$'s neighbors instead of all $N$ agents?
3. Can MADIFF be applied to general-sum settings or would it be limited to cooperative settings only?

**Questions:**

I hope to ask the authors' responses to my questions outlined in the weaknesses section.

---

> ### Author Response · Authors · 2023-11-21
>
> Dear reviewer, thanks for your sincere review and advice! Hereby we try to answer your questions.
>
> **Q1: "whether MADIFF can be applied to domains with high-dimensional inputs (e.g., image)"**
>
> **A1:**
> Thanks for your valuable suggestions! MADiff can be applied in environments with high-dimensional, e.g., image observations with more complicated encoder and decoder networks such as CNN or vision transformers. There are some successful cases of applying diffusion planning to single-agent and image-based tasks [1, 2], which can be incorporated into the MADiff framework. We leave this to future work and add a brief discussion to the conclusion of this paper.
>
> **Q2: "In Equation 9, inferring all other agents' observations could be very difficult when N is large. Could MADIFF scalability benefit by inferring only the agent i's neighbors instead of all N agents?"**
>
> **A2:**
> That is a good point, and your suggestion is valuable in boosting the scalability of MADiff. Selective opponent modeling is necessary in scaling to environments with very large N, and only inferring agent i's neighbors can reduce computation and modeling costs. However, neighbors defined under simple Euclidean distance may not be reasonable in some environments, while a learning-based distance metric would be more desirable. Another possible choice is to map the trajectories of all other agents to a low-dimensional embedding space and perform latent diffusion on it. Those modifications required are non-trivial, and we leave this direction as part of future work.
>
> **Q3: "Can MADIFF be applied to general-sum settings or would it be limited to cooperative settings only?"**
>
> **A3:**
> The MADiff framework can only be applied to cooperative tasks, which is the same as most prior works in offline MARL [3, 4]. Naive offline RL or generative modeling methods are likely to fail on non-cooperative tasks, since the optimal policy learned from static datasets can be exploited by other agents during evaluations if they change their policies to be different than behavior policies in datasets. Effective offline MARL in multi-player general-sum settings requires offline equilibrium finding (OEF), which is particularly challenging due to limited data support. Very little work has attempted to solve OEF and is limited to simple tasks with discrete state and action spaces [5].
>
> **References**
>
> [1] Chi, Cheng, et al. "Diffusion policy: Visuomotor policy learning via action diffusion." *arXiv preprint arXiv:2303.04137* (2023).
>
> [2] Du, Yilun, et al. "Learning universal policies via text-guided video generation." *Thirty-seventh Conference on Neural Information Processing Systems*. 2023.
>
> [3] Pan, Ling, et al. "Plan better amid conservatism: Offline multi-agent reinforcement learning with actor rectification." *International Conference on Machine Learning*. PMLR, 2022.
>
> [4] Yang, Yiqin, et al. "Believe what you see: Implicit constraint approach for offline multi-agent reinforcement learning." *Advances in Neural Information Processing Systems* 34 (2021): 10299-10312.
>
> [5] Li, Shuxin, et al. "Offline equilibrium finding." *arXiv preprint arXiv:2207.05285* (2022).

---

> > ### Comment · Reviewer_nDb9 · 2023-11-23
> > **Response to Rebuttal**
> >
> > I appreciate the authors for their detailed response to my feedback. The rebuttal addresses my questions (Q1-3). However, I agree with Reviewer Khwp's and QCKf's concerns, wherein some readers may interpret MADiff as Decision Diffuser (Ajay et al. 2023) with an additional attention layer based on the current writing. At the same time, I (partially) agree with the authors' points: 1. conditional generative modeling of multi-agent trajectories and 2. flexibility to handle various tasks (CTCE, CTDE, MATP). I would like to maintain my score but suggest to the authors to write more clearly, distinguishing the differences and importance of the contributions in a revised paper.

---

> > > ### Author Response · Authors · 2023-11-23
> > >
> > > Thank you again for thoughtful feedback and for dedicating time to review our paper.
> > >
> > > We greatly appreciate your confirmation on the findings and contributions of our work. We are preparing a revised manuscript to more prominently emphasize the unique contributions made by MADiff.

---

### Official Review · Reviewer_Khwp · 2023-11-07

**Soundness:** 3 good
**Presentation:** 3 good
**Contribution:** 3 good
**Rating:** 5
**Confidence:** 3

**Summary:**

This paper presents MADIFF, a centralized training-decentralized execution diffusion framework for multi-agent RL problems. MADIFF performs return-conditioned trajectory modeling with an attention-based diffusion architecture for information interchange among agents. MADIFF can be applied to both centralized and decentralized execution settings. Especially in the decentralized execution, MADIFF performs the opponent modeling, predicting the other agents' joint observations based on its local observation. In the experiments, MADIFF generally outperforms the baselines in MPE and is competitive in SMAC. For the MATP problem, MADIFF-C significantly outperforms the baseline.

**Strengths:**

1. A Diffusion-based offline MARL algorithm (an extension of DecisionDiffuser to a multi-agent setting) is presented, with a suitable attention mechanism for information interchange among agents in MARL.
2. In the experiments, MADIFF generally outperforms the baselines. The ablation study confirms that the proposed attention modules were indeed helpful.

**Weaknesses:**

1. The novelty of the work seems a bit limited. MADIFF heavily relies on the existing work DecisionDiffuser (Ajay et al., 2023), and it can be seen as its simple extension to the MARL setting, with an additional attention layer that processes information exchange among agents. Adopting an attention mechanism for multi-agents itself does not seem to be a new idea. (e.g. [1,2])
2. More baseline (MADT-KD [3]) could have been compared in the experiments.

[1] Iqbal et al., Actor-Attention-Critic for Multi-Agent Reinforcement Learning, 2019
[2] Wen et al., Multi-Agent Reinforcement Learning is A Sequence Modeling Problem, 2022
[3] Tseng et al., Offline Multi-Agent Reinforcement Learning with Knowledge Distillation, 2022

**Questions:**

1. In the top-middle of Figure 2, why is the planned trajectory of the planning agent (purple) different from the real sampled trajectory?
2. MADIFF relies on the inverse-dynamics model (IDM) for action selection, but it can be suboptimal when the transition dynamics are stochastic. In MARL, even though the underlying environment is deterministic, if we see the other agents as a part of the environment (thinking of decentralized execution), the transition dynamics (by other agents' actions) will be stochastic when other agents' policies are stochastic. Even in this situation, doesn't using IDM cause any problems?
3. If we view the MARL problem as a single-agent problem (by viewing joint action space as a single-agent large action space & joint observation space as a single-agent observation space), how is MADIFF-C different from DecisionDiffuser? Is MADIFF-C still performing better than DecisionDiffuser which operates in the joint observation/action space?

---

> ### Author Response · Authors · 2023-11-21
>
> Dear reviewer, thanks for your time reviewing this paper! We have conducted several additional experiments, and try to ease your concern below.  Hope we can ease your concerns.
>
> **Q1: "The novelty of the work seems a bit limited. MADIFF heavily relies on the existing work DecisionDiffuser (Ajay et al., 2023), and it can be seen as its simple extension to the MARL setting ... Adopting an attention mechanism for multi-agents itself does not seem to be a new idea."**
>
> **A1:** It is indeed that many successful design choices of [1] have inspired us and guided us to derive our method. Similarly, much of [1] also referred to the design of diffuser [2], but that doesn't take away from the fact that it is an excellent work.
>
> Compared with DM works in single-agent learning, MADiff is naturally designed for multi-agent problems. By using different conditioning during evaluation, the same framework can handle multi-agent coordination, opponent/teammate modeling, and joint trajectory prediction, which is supported by strong experimental results. We believe that the references in the single-agent learning domain do not diminish the novelty of our approach in the context of multi-agent learning.
>
> We acknowledge that the attention mechanism has been used for years in MAL, and we verified that it is also effective in conditional generative modeling of multi-agent trajectories.
>
> **Q2: "More baseline (MADT-KD) could have been compared in the experiments."**
>
> **A2:** Thank you for your suggestion! To our knowledge, MADT-KD does not have an open-sourced implementation, and it requires non-trivial changes on top of MADT's training code. We include MADT [3] as an additional baseline on SMAC datasets, and the results are pasted below:
>
> | Dataset     | MADiff-D       | MADiff-C       | MADT           |
> | :---------- | :------------- | :------------- | :------------- |
> | 3m-Good     | 18.8 $\pm$ 0.2 | 19.7 $\pm$ 0.1 | 19.0 $\pm$ 0.3 |
> | 3m-Medium   | 17.2 $\pm$ 0.3 | 18.4 $\pm$ 0.2 | 15.8 $\pm$ 0.5 |
> | 3m-Poor     | 11.2 $\pm$ 0.1 | 11.8 $\pm$ 1.0 | 4.2 $\pm$ 0.1  |
> | 5m6m-Good   | 16.5 $\pm$ 0.3 | 18.1 $\pm$ 0.1 | 16.8 $\pm$ 0.1 |
> | 5m6m-Medium | 16.3 $\pm$ 0.1 | 17.6 $\pm$ 0.4 | 16.1 $\pm$ 0.2 |
> | 5m6m-Poor   | 10.3 $\pm$ 0.5 | 11.0 $\pm$ 0.3 | 7.6 $\pm$ 0.3  |
>
> We also updated Table 1 to cover those results.
>
> **Q3: "In the top-middle of Figure 2, why is the planned trajectory of the planning agent (purple) different from the real sampled trajectory?"**
>
> **A3:** The plans on the top row are done at the early stage of an episode, and decisions are made in a decentralized way. Therefore, the purple agent at that time did not have much information about other agents' intentions. Then the purple agent may make conflicting plans with the red one, i.e., both want to cover the same landmark in the middle.
>
> As the purple agent gathers more information during execution, it identifies the red agent's inclination toward the middle landmark. Thus, the purple agent's opponent U-Net model tends to generate a future trajectory for the red agent, which arrives at the middle landmark. And because of the attention mechanism, the purple agent made corrections when generating their own plans (bottom-middle subfigure). The plans of all the agents in the bottom-middle row become conflict-free, and the conflicts are eliminated in real sampled trajectories.
>
> We have revised the explanation in Section 5.4 to make it clearer.

---

> > ### Author Response · Authors · 2023-11-21
> >
> > **Q4: "MADIFF relies on the inverse-dynamics model (IDM) for action selection, but it can be suboptimal when the transition dynamics are stochastic. In MARL, ..., the transition dynamics (by other agents' actions) will be stochastic when other agents' policies are stochastic. Even in this situation, doesn't using IDM cause any problems?"**
> >
> > **A4:** Thanks for your good question! We will separately discuss using IDM in stochastic transitions and transitions dependent on other agents' actions.
> >
> > First, we have to note that the stochastic state transition function $p(o'|o, a)$ does not necessarily lead to a stochastic inverse dynamics transition $p(a|o, o')$. Most of our tested environments do not exhibit high stochasticity in inverse dynamics, which can be effectively fit by state-conditioned Dirac distributions in practice. For potential environments with highly stochastic inverse dynamics, we can modify the IDM to output parametrized Gaussian distributions, which is similar to state-to-action policies used in stochastic environments.
> >
> > Indeed, in multi-agent environments, the transition dynamics might depend on other agents' actions, which can be stochastic. However, as we mentioned before, the IDM is only affected by the stochasticity of $p(a|o,o')$ rather than $p(o'|o,a)$. The objective of IDM is as simple as providing the feasible action given the current observation and the next observation predicted by the diffusion model, and the diffusion model is good at modeling stochastic distributions. The effect of other stochastic agents is already encoded in the generated next observation, which is the **input** of IDM. Therefore, such stochasticity does not make the learning objective of IDM much harder.
> >
> > **Q5: "If we view the MARL problem as a single-agent problem (...), how is MADIFF-C different from DecisionDiffuser? Is MADIFF-C still performing better than DecisionDiffuser which operates in the joint observation/action space?"**
> >
> > **A5:** If centralized control is allowed, the differences between MADiff and DecisionDiffuser operated on the joint state space are mainly in the model architecture. A naive approach to transforming multi-agent problems to fit into the DecisionDiffuser framework is concatenating all agents' observations together. MADiff adopts a more efficient way to model interactions between agents, where each agent's trajectory is processed by a single-agent U-Net, and self-attentions on agents' latents are introduced in decoder layers.
> >
> > We argue that the advantages of using attention modules can be explained in two aspects:
> >
> > + First, using the attention mechanism to model multi-agent interactions has been proven to be more effective than concatenation in prior MARL studies [3, 4]. To test whether such superiority also applies in diffusion modeling, we benchmark the model variant that uses a single U-Net to process concatenated multi-agent observations without using attention, which is denoted as ConcatDiff. We compare ConcatDiff to MADiff-C on two MPE spread datasets, and the results are summarized below.
> >
> >   | Dataset          | ConcatDiff      | MADiff-C            |
> >   | ---------------- | --------------- | ------------------- |
> >   | Spread-Expert    | 112.7 $\pm$ 4.3 | **114.7 $\pm$ 5.3** |
> >   | Spread-Md-Replay | 32.2 $\pm$ 3.5  | **47.2 $\pm$ 6.6**  |
> >
> >   We can see that MADiff-C outperforms ConcatDiff in those datasets thanks to a better model design.
> >
> > + Second, some MARL tasks require index-free control on a set of homogeneous agents [5], i.e., there is no predefined order of agents. Using attention without positional encodings to model multi-agent interactions can naturally handle the index-free setting, while the concatenation operation is ordered and requires additional designs to fit in those tasks.
> >
> > **References**
> >
> > [1] Ajay, Anurag, et al. "Is Conditional Generative Modeling all you need for Decision Making?." *The Eleventh International Conference on Learning Representations*. 2022.
> >
> > [2] Janner, Michael, et al. "Planning with Diffusion for Flexible Behavior Synthesis." *International Conference on Machine Learning*. PMLR, 2022.
> >
> > [3] Wen, Muning, et al. "Multi-agent reinforcement learning is a sequence modeling problem." *Advances in Neural Information Processing Systems* 35 (2022): 16509-16521.
> >
> > [4] Iqbal, Shariq, and Fei Sha. "Actor-attention-critic for multi-agent reinforcement learning." *International conference on machine learning*. PMLR, 2019.
> >
> > [5] Kingston, Peter, and Magnus Egerstedt. "Index-free multi-agent systems: An eulerian approach." *IFAC Proceedings Volumes* 43.19 (2010): 215-220.

---

> > > ### Author Response · Authors · 2023-11-23
> > >
> > > Dear Reviewer Khwp,
> > >
> > > We deeply appreciate the valuable and constructive feedback you provided. As the rebuttal discussion between reviewers and authors is coming to a close on November 23rd, we would greatly appreciate it if you could clarify any remaining concerns you may have. This will ensure that we can adequately address them during this period.
> > >
> > > Thank you for your attention to this matter. We are looking forward to your response.
> > >
> > > Best regards,
> > >
> > > The Authors

---

### Official Review · Reviewer_Gpw8 · 2023-11-07

**Soundness:** 2 fair
**Presentation:** 2 fair
**Contribution:** 2 fair
**Rating:** 6
**Confidence:** 3

**Summary:**

This paper extends previous work in diffusion-based offline RL to offline cooperative MARL, and in particular to the CTDE and centralized control settings.

The main contribution is the inclusion of an attention module in the diffusion model to help integrate information from other agents during centralized training and help coordination. The method is called MADiff

During online centralized control, an action is selected at each time step by first generating a future joint-observation sequence conditioned on the current joint-observation. An inverse dynamics model then infers the joint-action required to produce that observation sequence, and that is the joint-action taken by the agents.

The paper then evaluates MADiff in two offline MARL settings: SMAC (2 maps) and MPE (3 envs), and on the NBA trajectory prediction dataset.

**Strengths:**

This paper extends the application of diffusion models to offline multi-agent RL, with a non-trivial modification to the the usual U-Net architecture to facilitate coordination. The problem setting is significant, with potential applications to sports, urban planning, traffic prediction, ecology and other fields.

The empirical evaluation is relatively diverse, and compares to multiple relevant baselines.

At a high level, the paper is also generally well structured, and it is easy to follow from one section to the next. The Preliminaries section is particularly well contained.

**Weaknesses:**

**Clarity**
- While the paper is well structured, the writing could benefit from spell-checking and rephrasing since it often hurts understanding. A few examples:
  - "more high-frequency and less smooth nature of actions" : What does it mean for actions to be "high-frequency"?
  - the diffusing process is said to condition on information $y(\tau)$, which based on Fig. 1 includes "Returns" and "Other information". In that case, why are there multiple returns? Is it expected future discounted returns, or final returns for that trajectory? How are the returns encoded? What is the extra information?
  - reading section 3.1, it is unclear how the attention mechanism incorporates the information of other agents, and how this is supposed to help coordination.
  - the term "opponent" or "opponent modelling" is used repeatedly, despite the work only dealing with cooperative settings
  - Section 5.4 as a whole is unclear to me, including exactly what the Valid Ratio is measuring.

**Results**
- In SMAC, it is common to report the win rate rather than the return, because the return is much less interpretable. I would like the authors to provide the win rates.
- Figure 3 has no errorbars.
- No mention of the number of seeds used for Table 1 or Figure 3.
- The results for SMAC seem underwhelming, especially since SMAC lacks relevant stochasticity and partial observability

**Experiment Limitations**
- While the use of 2 different MARL settings and of the NBA dataset is welcome, the paper does not evaluate MADiff in settings where coordination is particularly challenging, either due to stochasticity, partial observability or both. It also does not acknowledge any potential limitations of the method in such settings. For instance, I encourage the authors to consider environments such as SMACv2 [1], Multi-Agent Mujoco [2] or Hanabi [3].


- [1] https://arxiv.org/abs/2212.07489
- [2] https://github.com/schroederdewitt/multiagent_mujoco
- [3] https://arxiv.org/abs/1902.00506

**Questions:**

1. Why downsample the NBA dataset from 25 Hz to 5 Hz? How does this affect MADiff and the baseline?

2. Have the authors tried training MADiff on combined datasets of mixed quality (e.g. Good+Medium+Poor)? I suspect the additional data would benefit MADiff the most by improving the inverse dynamics model without hurting the quality of the generated trajectories when conditioning on high return.

3. The environment with the most agents is *5m6m*, which has only 5 agents. How does MADiff scale to a large number of agents?

---

> ### Author Response · Authors · 2023-11-21
>
> Dear reviewer, thanks for your time! We have conducted additional experiments on Multi-Agent Mujoco datasets and hereby answer your questions below. Hope we can ease your concerns.
>
> **Q1: "What does it mean for actions to be 'high-frequency'?"**
>
> **A1:**
> We are sorry for the confusion. The term "frequency" refers to using the Fourier transform to transform the state or action sequences in the time domain to the frequency domain. Action sequences are less smooth across time steps and thus have a lot of high-frequency components after transformation.
>
> To make it clearer, we have removed the statement about action frequency.
>
> **Q2: About "Returns" and "Other information"  in Figure 1.**
>
> **A2:**
> We are sorry for the confusion. During training, "Returns" in Fig. 1 is set to the cumulative discounted reward starting from time step $t$. In some environments, agents can have different returns at some time step, e.g., penalties on the specific agent that collides with others in MPE, thus we use the plural form to denote returns for each agent. During evaluation, we set returns to the same high-enough value for all agents. Before input to the U-Net residual blocks, Returns are passed through an MLP and concatenated with the diffusion time step embedding.
>
> "Other information" is task-related condition variables other than returns. For example, in the NBA dataset, we need player embeddings and ball positions as inputs. "Other information" is passed through a separate embedding network and concatenated with return embeddings. We have revised the caption of Fig. 1 to include descriptions of these two terms.
>
> **Q3: "it is unclear how the attention mechanism incorporates the information of other agents, and how this is supposed to help coordination"**
>
> **A3:**
> The attention mechanism helps coordination in both centralized control and decentralized execution settings.
> When centralized control is allowed, the attention module enables the diffusion model to jointly plan trajectories for all agents, which is essential in tasks that require coordination.
>
> In a decentralized execution setting, although different agents make decisions independently, they can use the same attention network to jointly infer other agents' future trajectories and plan their own trajectories. Since the attention network is trained in a centralized manner, the generated trajectories for all agents should be consistent. Intuitively, the consistency constraint requires the ego agent to first think at a higher level, i.e., make a coordinated plan for all agents, and then place itself in that plan. The coordination among agents can be improved by performing "imagined" planning for others (i.e., opponent modeling).
>
> **Q4: "the term "opponent" or "opponent modelling" is used repeatedly, despite the work only dealing with cooperative settings."**
>
> **A4:**
> Thank you for pointing this out. We use the term "opponent" to refer to other teammates in cooperative games, which is also adopted in some prior works [1, 2]. Since most methods in modeling adversary agents in competitive games can be used in modeling teammates, using the same term "opponent" can ensure better coverage of related works.
>
> **Q5: "Section 5.4 as a whole is unclear to me, including exactly what the Valid Ratio is measuring."**
>
> **A5:** We are sorry for the confusion and have rewritten Section 5.4 to make it more clear.
>
> **Q6: "In SMAC, it is common to report the win rate ... would like the authors to provide the win rates".**
>
> **A6:**
> Most prior offline MARL works use self-collected datasets, which makes it hard to measure the progress in this field. Instead, we use the off-the-grid MARL dataset [3], which is open-sourced and released with comprehensive baseline results. For SMAC tasks, they did not report the win rates but only the scores of baseline algorithms. As a result, we do not compare the win rates of MADiff in our experiments. We want to highlight that there are some offline MARL works [4, 5] that also use the average score as the only metric for SMAC.
>
> To make the readers better understand the magnitude of scores in SMAC, we have added Fig. 5 in the supplementary material to demonstrate the distribution of scores in each SMAC dataset we used.
>
> **Q7: Issues concerning Figure 3 and Table 1.**
>
> **A7:**
> Thank you for your careful check! Results are computed with 3 different seeds. In the revised paper, we mention the number of seeds in the captions of Fig. 3 and Tab. 1. The error bar is added to Fig. 3.
>
> **Q8: "The results for SMAC seem underwhelming, especially since SMAC lacks relevant stochasticity and partial observability."**
>
> **A8:**
> To our knowledge, each agent in SMACv1 can only receive local observations within a fixed sight range [6], which makes the environment partially observable. Although SMACv1 lacks relevant stochasticity, we highlight that both MA-ICQ and MA-CQL are strong baselines in offline MARL, and MADiff achieves competitive results.

---

> > ### Author Response · Authors · 2023-11-21
> >
> > **Q9: "the paper does not evaluate MADiff in settings where coordination is particularly challenging, either due to stochasticity, partial observability or both."**
> >
> > **A9:**
> > Thanks for your good question, but you might misunderstand our experiment settings. Although we did not consider environments with both stochasticity and partial observability, either of them alone was present in our experiments. In particular, partial observability exists in SMAC due to limited sight range, and MPE has stochasticity in initial state distributions. Moreover, since we consider offline multi-agent policy learning from static datasets, it is challenging for agents to effectively coordinate during online evaluations.
> >
> > **Q10: "For instance, I encourage the authors to consider environments such as SMACv2, Multi-Agent Mujoco or Hanabi."**
> >
> > **A10:**
> > Thanks for your suggestions! We evaluate MADiff on Multi-agent Mujoco datasets in off-the-grid MARL and results are presented as follows:
> >
> > | Dataset             | BC         | TD3+BC | OMAR | MADiff-D |
> > | ------------------- | ---------- | ------ | ---- | -------- |
> > | 2halfcheetah-Good   | 6846 $\pm$ 574 | 7025 $\pm$ 439 | 1434 $\pm$ 1903 | **8254 $\pm$ 179** |
> > | 2halfcheetah-Medium | 1627 $\pm$ 187 | **2561 $\pm$ 82** | 1892 $\pm$ 220 | 2215 $\pm$ 27 |
> > | 2halfcheetah-Poor   | 465 $\pm$ 59 | 736 $\pm$ 72 | 384 $\pm$ 420 | **751 $\pm$ 14** |
> > | 4ant-Good           | 2802 $\pm$ 133 | 2628 $\pm$ 971 | 344 $\pm$ 631 | **3090 $\pm$ 26** |
> > | 4ant-Medium         | 1617 $\pm$ 153 | **1843 $\pm$ 494** | 929 $\pm$ 349 | 1679 $\pm$ 93 |
> > | 4ant-Poor           | 1033 $\pm$ 122 | 1075 $\pm$ 96 | 518 $\pm$ 112 | **1268 $\pm$ 51** |
> >
> > **Q11: "Why downsample the NBA dataset from 25 Hz to 5 Hz? How does this affect MADiff and the baseline?"**
> >
> > **A11:**
> > The downsampling is adopted by the baseline method [7] for NBA dataset, and we use the same data processing for proper comparisons. We suspect the reason for downsampling is that most consecutive states have too small differences under the original 25Hz. This can make the sequences lengthy and contain a lot of noisy information, which has a negative impact on model learning.
> >
> > **Q12: "Have the authors tried training MADiff on combined datasets of mixed quality? "**
> >
> > **A12:**
> > The qualities of SMAC datasets are already mixed. The naming of datasets in off-the-grid MARL datasets (Good, Medium, Poor) does not mean that each of them is sampled by a single policy but indicates the overall quality of the dataset. In fact, each dataset is collected by three sets of independently trained policies with different levels of expertise [3]. We have included Fig. 5 in the Appendix to show the return distributions of each dataset, and briefly describe how those datasets are collected.
> >
> > **Q13: "How does MADiff scale to a large number of agents?"**
> >
> > **A13:**
> > As mentioned in Section 5.5, we choose to use a shared U-Net model for all agents, where different agents' data can be batched together and passed through the network. It does not cost much more inference time as the number of agents increases with GPU-accelerated computing. For self-attention among agents' embeddings, it can also scale with linear complexity [8].
> >
> > When the number of agents is extremely large, we can modify the opponent/teammate modeling to operate in a projected low-dimensional space with a latent diffusion model [9], thus avoiding the curse of dimensionality. We leave this direction to future work.

---

> > > ### Author Response · Authors · 2023-11-21
> > >
> > > **References**
> > >
> > > [1] Yu, Xiaopeng, et al. "Model-based opponent modeling." *Advances in Neural Information Processing Systems* 35 (2022): 28208-28221.
> > >
> > > [2] Zhang, Weinan, et al. "Model-based Multi-agent Policy Optimization with Adaptive Opponent-wise Rollouts." *Proceedings of the Thirtieth International Joint Conference on Artificial Intelligence.* 2021.
> > >
> > > [3] Formanek, Claude, et al. "Off-the-Grid MARL: Datasets and Baselines for Offline Multi-Agent Reinforcement Learning." *Proceedings of the 2023 International Conference on Autonomous Agents and Multiagent Systems*. 2023.
> > >
> > > [4] Tseng, Wei-Cheng, et al. "Offline Multi-Agent Reinforcement Learning with Knowledge Distillation." *Advances in Neural Information Processing Systems* 35 (2022): 226-237.
> > >
> > > [5] Yang, Yiqin, et al. "Believe what you see: Implicit constraint approach for offline multi-agent reinforcement learning." *Advances in Neural Information Processing Systems* 34 (2021): 10299-10312.
> > >
> > > [6] Whiteson, S., et al. "The StarCraft multi-agent challenge." *Proceedings of the International Joint Conference on Autonomous Agents and Multiagent Systems, AAMAS*. 2019.
> > >
> > > [7] Alcorn, Michael A., and Anh Nguyen. "baller2vec++: A look-ahead multi-entity transformer for modeling coordinated agents." *arXiv preprint arXiv:2104.11980* (2021).
> > >
> > > [8] Shen, Zhuoran, et al. "Efficient attention: Attention with linear complexities." *Proceedings of the IEEE/CVF winter conference on applications of computer vision*. 2021.
> > >
> > > [9] Rombach, Robin, et al. "High-resolution image synthesis with latent diffusion models." *Proceedings of the IEEE/CVF conference on computer vision and pattern recognition*. 2022.

---

> > > > ### Comment · Reviewer_Gpw8 · 2023-11-22
> > > > **Reviewer Follow-up**
> > > >
> > > > I think their authors for their thorough response. They have addressed some of my concerns, but I still have some below.
> > > >
> > > > **Q2: Returns and Other information**
> > > >
> > > > I still find there is too little information for something potentially crucial to the performance of MADiff. Regarding returns, if I understand correctly, you mention utilizing a factorized returns for agents (e.g. collision penalty, reward for reaching the goal, etc.). This is already problematic, because splitting the reward function into different components and giving the components separately to the policy is non-standard, as it implies domain knowledge about the environment. Do you also perform this factorization for your baselines? If no, then it is a potential boost to MADiff.
> > > >
> > > > Crucially, the authors say _"agents can have different returns at some time step"_. This is true, but typically those rewards are accrued into a single global reward shared between all agents (as the paper says in section 2.1). This is why credit assignment is required in cooperative MARL and why CTDE is so hard.
> > > >
> > > > To summarize, any form of reward factorization, and especially reward factorization per agent should be a) clearly disclosed in the paper and b) clearly justified, because it simplifies the CTDE problem greatly.
> > > >
> > > > Similarly, any "Other information" provided to the method should be clearly disclosed on a per-environment basis, to make it clear whether or not the other information contains domain knowledge which gives an advantage to MADiff over the baselines. For instance, what is the extra information in SMAC?
> > > >
> > > > **Q4: "Opponent modelling"**
> > > >
> > > > The definition of "opponent" is "someone who competes with or opposes another in a contest, game, or argument", so using it to refer to teammates is confusing, regardless of prior usage in the literature. This is even worse in SMAC, where there are teammates and enemies (i.e. opponents), even if the opponents are controlled by a fixed policy. I suggest the authors simply use "teammate" instead of "opponent", and maintain the related works coverage as is.
> > > >
> > > > **Q5: Section 5.4**
> > > >
> > > > Thank you for rewriting. This is now clearer. However, there is a mistake in Fig. 2 (bottom left). All three trajectory predictions are from the point of view of Red agent. You do not show the predictions for the Green or Blue agent.
> > > >
> > > > **Q6: Win rates**
> > > >
> > > > Even though past work in offline MARL has omitted win rates in SMAC, I encourage the authors to add them to bridge the gap between the online and offline literatures. They should do so even if they cannot evaluate all baselines to get the win rates.
> > > >
> > > > **Q8: Partial observability in SMAC**
> > > >
> > > > Yes, SMACv1 agents have a limited sight range, but that does not make it relevant partial observability. The SMACv2 demonstrates that it is possible to learn a strong policy in SMACv1 maps by conditioning *only on the timestep and nothing else*.  It is okay to have a method that only works well in environments without meaningful partial observability as it can still have many applications, but those limitations need to be explicit.
> > > >
> > > > **Q9: Stochasticity and Partial Observability**
> > > >
> > > > As mentioned above, SMACv1 partial observability is very limited. MPE has stochasticity in the initial states, but once those are observed, the rest of the dynamics are deterministic. Furthermore, partial observability and stochasticity cannot only be considered in isolation. For a discussion on the interplay between the two, please refer to section 4 of SMACv2 [1].
> > > >
> > > > **Q10: Multi-agent Mujoco**
> > > >
> > > > I thank the authors for the additional experiment! To correctly interpret the results, however, one needs details on the settings used. I assume 2halfcheetah and 4ant refer to the 2-agent halfcheetah and 4-agent ant default configurations from the repo. Is that correct? Do you provide any "Other information" to the diffusion model, and if so, what?
> > > >
> > > > Also, you should add the results to the revised paper.
> > > >
> > > > Given my remaining concerns, I will maintain my score for now.
> > > >
> > > >
> > > > [1] https://arxiv.org/pdf/2212.07489.pdf

---

> > > > > ### Author Response · Authors · 2023-11-23
> > > > > **Responses to remaining concerns**
> > > > >
> > > > > Thank you for the prompt reply! We are glad you acknowledge our responses, and we try to ease your remaining concerns in the following part.
> > > > >
> > > > > **Q2: "Returns and Other information"**
> > > > >
> > > > > **A2:** We are sorry for not fully addressing your concern in prior responses. Our statements in former **A2** may be confusing, and we have to clarify that **no additional information or domain knowledges are used in MADiff compared to baseline methods**.
> > > > >
> > > > > **Other information:**
> > > > >
> > > > > The other information is **used in the NBA dataset only**, which is why we marked "Other Information" in Fig. 1 as semi-transparent. The MATP task requires predicting players' future trajectories given players' and ball's historical trajectories, player IDs, and a binary variable indicating the side of each player's frontcourt (i.e., the direction of their team's hoop). We refer to inputs other than players' historical trajectories as "other information." The baseline method, Baller2Vec++ [1], takes embeddings of other information as input tokens to their transformer model. We concatenate the embeddings of other information with diffusion time embeddings as side inputs to each U-Net block.
> > > > >
> > > > > In order to avoid confusion in offline MARL experiments, we remove "Other Information" from Fig.1 and describe how other information in the NBA dataset is encoded in Section 5.1.
> > > > >
> > > > > **Returns:**
> > > > >
> > > > > For returns input to the model, we clarify that **we did not do extra reward factorization than baseline methods**. All agents receive the same reward for SMAC and Multi-agent Mujoco experiments at each time step. During training, the same cumulative team return is simply repeated and input to each agent's U-Net.
> > > > >
> > > > > For MPE experiments, we use datasets and a fork of the environment provided by OMAR [2]. They seem to be using an earlier version of MPE where agents can receive different rewards.
> > > > > For example, in the Spread task, team reward is defined using the distance of each landmark to its closest agent, which is the same for all agents.
> > > > > However, when an agent collides with others, it will receive the team reward minus a penalty term (https://github.com/ling-pan/OMAR/blob/0e52ad6fc23585a83eb771e9315d4966e3faa128/multiagent-particle-envs/multiagent/scenarios/simple_spread.py#L115-L116). The collision reward has been brought into the team reward in the official repository since this commit (https://github.com/openai/multiagent-particle-envs/commit/6ed7cac026f0eb345d4c20232bafa1dc951c68e7). However, the fork provided by OMAR still uses the legacy version (https://github.com/ling-pan/OMAR/blame/0e52ad6fc23585a83eb771e9315d4966e3faa128/multiagent-particle-envs/multiagent/environment.py#L36). To make a fair and proper comparison, we use the OMAR's dataset and environment where all baselines are trained and evaluated. We have to note that different rewards for agents only happen at **very few steps**, which might not contradict the fully cooperative setting much. For example, OMAR's expert split of the Spread dataset consists of 1M steps, and different rewards are recorded only at less than 1.5% (14929) steps.
> > > > >
> > > > > We sincerely appreciate the discussion with you, and have included the clarification on MPE datasets in Appendix C.1.
> > > > >
> > > > > **Q4: "Opponent modelling"**
> > > > >
> > > > > **A4:** Thank you for the suggestion. We agree that using "opponent" is confusing, especially in SMAC tasks. In the revised manuscript, we renamed all "opponent" to "teammate" when describing our method.
> > > > >
> > > > > **Q5: Section 5.4**
> > > > >
> > > > > **A5:** Thank you for careful checking! We apologize for the mistake and have replaced Fig. 2 with the correct figure.
> > > > >
> > > > > **Q6: Win rates**
> > > > >
> > > > > **A6:** Thank you for your valuable suggestion! We agree that reporting win rates can provide more information on comparisons with online results. We will reevaluate our models and rerun baseline algorithms with implementations provided by the off-the-grid dataset to record win rates for comparisons. Results will be reported in the Appendix once they are available.
> > > > >
> > > > > **Q8 & Q9: SMACv2 Environments**
> > > > >
> > > > > **A8 & 9:** Thank you for the valuable discussions, and we now fully understand the importance of evaluating on SMACv2 datasets. We are running experiments on datasets of zerg_5_vs_5 and terran_10_vs_10 from SMACv2 and will update the results in the manuscript after they are available. Before that, we have discussed the current limitations of model scalability and experiment validations in Section 6.

---

> ### Author Response · Authors · 2023-11-23
> **Responses to remaining concerns cont.**
>
> **Q10: Multi-agent Mujoco**
>
> **A10:** We are delighted that you acknowledge our added experiments. The naming of different Multi-agent Mujoco tasks follows the off-the-grid datasets (https://github.com/instadeepai/og-marl), where 2halfcheetah and 4ant refer to the 2-agent halfcheetah and 4-agent. The environment setting is the same as the default configurations from the repo of the Multi-agent Mujoco environment (https://github.com/schroederdewitt/multiagent_mujoco). As we mentioned in **A2**, no other information is used in Multi-agent Mujoco experiments. Results on Multi-agent Mujoco are added to the revised paper, accompanying necessary descriptions.
>
>
> **References**
>
> [1] Alcorn, Michael A., and Anh Nguyen. "baller2vec++: A look-ahead multi-entity transformer for modeling coordinated agents." *arXiv preprint arXiv:2104.11980* (2021).
>
> [2] Pan, Ling, et al. "Plan better amid conservatism: Offline multi-agent reinforcement learning with actor rectification." *International Conference on Machine Learning*. PMLR, 2022.

---

### Author Response · Authors · 2023-11-21
**Response to all the reviewers**

We want to express our sincere gratitude to all the reviewers for their valuable feedback and insightful comments.

We have updated a revised manuscript, and some important changes are made:

1. **Added more information on SMAC&MPE datasets** in Appendix C. We discuss an issue in MPE dataset, which makes it slightly contradict the fully cooperative setting. We also briefly describe how SMAC datasets are collected and include violin plots of the return/score distribution of different datasets.
2. **Added necessary information on implementations of baseline algorithms** in Appendix B.
3. **Added related works on opponent modeling** in Section 4.
4. **Added MADT results on SMAC datasets** in Table 1.
5. **Added results on Multi-agent Mujoco datasets** in Table 1.
6. **Added discussions of limitations** in the conclusion section regarding model scalability and experiment validations.
7. **Rewritten the qualitative analysis on opponent modeling** in Section 5.4.
8. **Renamed "opponent modeling" to "teammate modeling"** to avoid ambiguity since we do not model adversary agents.

In the following responses, we address each reviewer's comments and provide detailed explanations. We are willing to engage in further discussion regarding specific issues.

---

> ### Author Response · Authors · 2023-11-21
> **New experiment results**
>
> We have conducted addtional experiments to address reviewers' questions, and new results are summarized as follows:
>
> 1. We benchmark MADiff on Multi-agent Mujoco datasets:
>
>    | Dataset             | BC             | TD3+BC             | OMAR            | MADiff-D           |
>    | ------------------- | -------------- | ------------------ | --------------- | ------------------ |
>    | 2halfcheetah-Good   | 6846 $\pm$ 574 | 7025 $\pm$ 439     | 1434 $\pm$ 1903 | **8254 $\pm$ 179** |
>    | 2halfcheetah-Medium | 1627 $\pm$ 187 | **2561 $\pm$ 82**  | 1892 $\pm$ 220  | 2215 $\pm$ 27      |
>    | 2halfcheetah-Poor   | 465 $\pm$ 59   | 736 $\pm$ 72       | 384 $\pm$ 420   | **751 $\pm$ 14**   |
>    | 4ant-Good           | 2802 $\pm$ 133 | 2628 $\pm$ 971     | 344 $\pm$ 631   | **3090 $\pm$ 26**  |
>    | 4ant-Medium         | 1617 $\pm$ 153 | **1843 $\pm$ 494** | 929 $\pm$ 349   | 1679 $\pm$ 93      |
>    | 4ant-Poor           | 1033 $\pm$ 122 | 1075 $\pm$ 96      | 518 $\pm$ 112   | **1268 $\pm$ 51**  |
>
> 2. We compare MADiff to ConcatDiff, a naive application of single-agent DM to MAL which concatenates all agents' observations:
>
>    | Dataset          | OMAR            | ConcatDiff-D    | MADiff-D        | ConcatDiff-C | MADiff-C    |
>    | :--------------- | :-------------- | :-------------- | --------------- | ------------ | ----------- |
>    | Spread-Expert    | 114.9 $\pm$ 2.6 | 106.4 $\pm$ 3.5 | 98.4 $\pm$ 12.7 | 112.7 $\pm$ 4.3  | 114.7 $\pm$ 5.3 |
>    | Spread-Md-Replay | 37.9 $\pm$ 12.3 | 31.5 $\pm$ 2.7  | 42.9 $\pm$ 11.6 | 32.2 $\pm$ 3.5   | 47.2 $\pm$ 6.6  |
>
> 3. We compare MADiff with MADT:
>
>    | Dataset     | MADiff-D       | MADiff-C       | MADT           |
>    | :---------- | :------------- | :------------- | :------------- |
>    | 3m-Good     | 18.8 $\pm$ 0.2 | 19.7 $\pm$ 0.1 | 19.0 $\pm$ 0.3 |
>    | 3m-Medium   | 17.2 $\pm$ 0.3 | 18.4 $\pm$ 0.2 | 15.8 $\pm$ 0.5 |
>    | 3m-Poor     | 11.2 $\pm$ 0.1 | 11.8 $\pm$ 1.0 | 4.2 $\pm$ 0.1  |
>    | 5m6m-Good   | 16.5 $\pm$ 0.3 | 18.1 $\pm$ 0.1 | 16.8 $\pm$ 0.1 |
>    | 5m6m-Medium | 16.3 $\pm$ 0.1 | 17.6 $\pm$ 0.4 | 16.1 $\pm$ 0.2 |
>    | 5m6m-Poor   | 10.3 $\pm$ 0.5 | 11.0 $\pm$ 0.3 | 7.6 $\pm$ 0.3  |

---

### Meta-Review · Area_Chair_HstY · 2023-12-07

**Metareview:**

This paper introduces a diffusion model for data augmentation in offline multi-agent RL.

Specifically, the strengths of the paper are as follows:
- relatively straightforward method
- diverse experimental evaluation

However, there are also some key weaknesses that were pointed out by the reviewers:
- limited novelty due to this being a direct extension of the diffusion based offline RL methods from single agent RL to the multi-agent setting. The only new part here is the attention.
- some key experiments are missing, e.g. reviewers requested evaluation on SMACv2, which was not provided.
- some of the reviewers struggled to properly follow the paper due to clarity issues.
- unclear if the baseline methods would also benefit from the attention architecture, which makes it more challenging to compare like-for-like

Based on the reviews and the current state of the paper I do not think that the paper is ready for acceptance in its current form.
I strongly recommend that the authors implement the changes suggested by the reviewers before submitting to the next conference.

**Justification For Why Not Higher Score:**

On balance, the weaknesses above outweigh the novelty of the method.

**Justification For Why Not Lower Score:**

N/A

---

### Decision · Program_Chairs · 2024-01-16

Reject